# HMGB1: A Central Node in Cancer Therapy Resistance

**DOI:** 10.3390/ijms262412010

**Published:** 2025-12-13

**Authors:** Bashar A. Alhasan, Boris A. Margulis, Irina V. Guzhova

**Affiliations:** Laboratory of Cell Protection Mechanisms, Institute of Cytology, Russian Academy of Sciences, Tikhoretsky Ave. 4, 194064 St. Petersburg, Russia; margulis@incras.ru (B.A.M.); irina.guzhova@incras.ru (I.V.G.)

**Keywords:** HMGB1, cancer therapy resistance, DNA damage repair, autophagy, apoptosis, ferroptosis, pyroptosis, multi drug resistance

## Abstract

Cancer therapy resistance emerges from highly integrated molecular systems that enable tumor cells to evade cell death and survive cytotoxic therapeutic stress. High Mobility Group Box 1 (HMGB1) is increasingly gaining recognition as a central coordinator of these resistance programs. This review delineates how HMGB1 functions as a molecular switch that dynamically redistributes between cellular compartments in response to stress, with each localization enabling a distinct layer of resistance. In the nucleus, HMGB1 enhances chromatin accessibility and facilitates the recruitment of DNA repair machinery, strengthening resistance to radio- and chemotherapeutic damage. Cytosolic HMGB1 drives pro-survival autophagy, maintains redox stability, and modulates multiple regulated cell death pathways, including apoptosis, ferroptosis, and necroptosis, thereby predominantly shifting cell-fate decisions toward survival under therapeutic pressure. Once released into the extracellular space, HMGB1 acts as a damage-associated molecular pattern (DAMP) that activates key pro-survival and inflammatory signaling pathways, establishing microenvironmental circuits that reinforce malignant progression and therapy escape. HMGB1 further intensifies resistance through upregulation of multidrug resistance transporters, amplifying drug efflux. Together, these compartmentalized functions position HMGB1 as a central node in the networks of cancer therapy resistance. Emerging HMGB1-targeted agents, ranging from peptides and small molecules to receptor antagonists and nanoformulations, show promise in reversing resistance, but clinical translation will require precise, context- and redox-informed HMGB1 targeting to overcome multifactorial resistance program in refractory cancers.

## 1. Introduction

Cancer therapy resistance remains one of the most formidable challenges in modern oncology, limiting the long-term efficacy of conventional, targeted, and immune therapies. Despite substantial advances in molecular-targeted combination therapies and drug delivery systems, the long-term efficacy of treatment is often compromised by the ability of tumor cells to adapt and survive under therapeutic pressure. Therapy resistance is a multifactorial adaptive process that emerges through a complex interplay of molecular and cellular adaptations, including activation of pro-survival while inhibiting cell death pathways, enhancement of DNA repair, metabolic reprogramming, epigenetic modifications, and the acquisition of stress-tolerant stem-like phenotypes [1,2]. These adaptive processes enable cancer cells not only to withstand cytotoxic insults but also to repopulate and drive disease recurrence and metastasis. Understanding the molecular regulators that coordinate such adaptive responses is therefore crucial for overcoming therapeutic resistance and improving patient outcomes [3,4,5].

Among the key modulators of cellular adaptation is HMGB1, a highly conserved, ubiquitous multifunctional protein that plays diverse roles in genome organization, stress signaling, and intercellular communication. HMGB1 (previously termed amphoterin) was first identified in calf thymus extracts in 1973 and named for its rapid electrophoretic mobility on polyacrylamide gels. It belongs to the HMGB family of non-histone chromatin-binding proteins (HMGB1-4) that primarily reside in the nucleus under basal conditions and function as DNA chaperones, regulating chromatin plasticity and facilitating various DNA-dependent processes, including replication, transcription, recombination, and repair. HMGB proteins are broadly expressed, and HMGB1 is the most abundant and widely studied member due to its striking functional versatility in both physiological and pathological contexts [6,7].

Structurally, HMGB1 consists of 215 amino acids organized into two positively charged DNA-binding domains, the A-box and B-box, followed by a negatively charged acidic tail at the C-terminus [8]. The A- and B-boxes enable HMGB1 to bind and bend DNA, thereby influencing chromatin structure and transcription factor accessibility, while the acidic tail modulates these interactions by competing for intramolecular binding sites. The protein contains two nuclear localization sequences (NLS1 and NLS2) that regulate its nuclear-cytoplasmic trafficking, which is tightly regulated by post-translational modifications (PTMs). HMGB1 also contains three redox-sensitive conserved cysteine residues, C23, C45, and C106, that determine the protein’s conformational state and extracellular activities [9,10]. Moreover, distinct amino acid motifs in HMGB1 mediate its binding to cell surface receptors, including residues 89–108, mediating binding to TLR4, and 150–183 residues for the receptor for advanced glycation end-products (RAGE), enabling HMGB1 to communicate stress signals beyond the cell [11,12]. This modular organization endows HMGB1 with exceptional plasticity, allowing it to function as a nuclear architectural protein, a cytosolic stress regulator, and an extracellular signaling molecule (Figure 1).

Under normal physiological conditions, HMGB1 resides predominantly in the nucleus, where it stabilizes nucleosomes, promotes DNA bending, and participates in chromatin remodeling. However, HMGB1 is highly responsive to cellular stress [13]. Various stimuli, including oxidative stress, hypoxia, nutrient deprivation, and DNA damage, can trigger its translocation from the nucleus to the cytoplasm [14]. This translocation is mediated by post-translational modifications such as acetylation, phosphorylation, methylation, and ADP-ribosylation, which neutralize the nuclear localization signals and promote export through the chromosome-region maintenance (CRM1) pathway [15]. Once in the cytoplasm, HMGB1 engages in multiple stress-response pathways, coordinating adaptive processes that implicate whether a cell survives or undergoes regulated cell death [16].

Beyond the intracellular compartment, HMGB1 can also be released into the extracellular space. This release occurs through two distinct routes: passive release from necrotic or dying cells and active secretion by stressed or activated cells. The active secretion of HMGB1 is unusual in that it bypasses the classical endoplasmic reticulum–Golgi pathway due to the absence of a signal peptide and instead relies on secretory lysosome exocytosis or autophagy-associated mechanisms [17,18]. Once outside the cell, HMGB1 acts as a damage-associated molecular pattern (DAMP) molecule that signals tissue stress and injury. Acting as a potent alarmin, HMGB1 activates innate immune receptors, promotes cytokine secretion, and orchestrates leukocyte recruitment, thereby bridging sterile tissue damage and immune activation. These immunomodulatory and inflammatory roles of extracellular HMGB1 have been extensively documented over the past two decades, particularly in the context of infection, sepsis, and inflammatory diseases [8,9,19,20]. In cancer, extracellular HMGB1 has also been implicated in shaping the tumor immune microenvironment by modulating antigen presentation, immune suppression, and response to immunotherapies [6,8,20].

The biological activity of extracellular HMGB1 is tightly governed by its redox state, which determines whether it promotes chemotaxis, inflammation, or immune tolerance. Extracellular HMGB1 can exist in reduced, disulfide, or fully oxidized forms, each conferring distinct biological activities. In the reduced (all-thiol) form, HMGB1 behaves primarily as a chemotactic mediator by forming a heterocomplex with CXCL12 and activating CXCR4-driven cell migration. The disulfide HMGB1 form (containing a C23-C45 bond with a reduced C106) binds to toll-like receptor 4 (TLR4) and triggers NF-κB–dependent cytokine release, thereby amplifying inflammatory signaling. Further oxidation of all three cysteines to the sulfonyl state generates terminally oxidized HMGB1, an inert form that lacks cytokine or chemotactic activity and is associated with immune tolerance and resolution of inflammation [18]. These redox-dependent transitions underscore that HMGB1’s extracellular functions are not uniform but dynamically shaped by oxidative stress, therapy-induced cell death, and the tumor microenvironment, influencing whether HMGB1 promotes antitumor immunity or contributes to resistance-associated inflammation. However, despite this well-established immunological perspective, important questions remain regarding the non-immune mechanisms through which HMGB1 contributes to tumor persistence and therapy resistance.

Over recent years, several comprehensive reviews have examined the multifunctional roles of HMGB1 in cancer, including its immunological functions, its impact on the tumor microenvironment, and its contributions to inflammation-driven tumor progression [6,8,9,19,20]. Other reviews have discussed selected aspects of HMGB1-mediated autophagy, DNA repair, apoptosis regulation, multidrug resistance, and its broader relevance from molecular mechanisms to clinical applications [9,10,12,16,20,21,22,23,24,25,26,27]. Although these important works have greatly shaped the field and advanced our understanding, most have highlighted HMGB1’s functions in specific biological contexts rather than considering how various mechanisms intersect to mutually drive therapeutic escape.

In parallel, accumulating evidence indicates that HMGB1 exerts profound effects on cancer cell survival that extend far beyond immune modulation, with intracellular and extracellular HMGB1 regulate stress-adaptive signaling networks that enable tumor cells to endure therapeutic insults [28,29]. Therefore, the novelty of the present review lies in synthesizing these diverse HMGB1 activities into a coherent and mechanistically integrated model of therapy resistance. Rather than treating HMGB1’s roles as separate phenomena, we highlight how its nuclear, cytosolic, and extracellular functions form a coordinated resistance network that enables cancer cells to withstand therapeutic stress. Specifically, we integrate four major domains of cancer therapy resistance, genome maintenance and DNA repair, stress-induced autophagy, cell death escape, and multidrug resistance (MDR) programs, and describe how their interplay with HMGB1 positions this protein as a central organizer linking these processes. By integrating emerging evidence across these fields, we aim to provide readers with a comprehensive perspective on HMGB1 as a primary mediator of therapy resistance and to outline opportunities for its pharmacological targeting in cancer treatment.

## 2. HMGB1 in Genome Maintenance and DNA Repair Mechanisms

### 2.1. HMGB1 in Double-Strand Break (DSB) Repair: A Modulator of Genome Integrity and Radio Resistance

HMGB1 has emerged as a pivotal regulator of DNA double-strand break repair mechanisms, including homologous recombination (HR) repair and non-homologous end joining (NHEJ) [20,30]. Elevated HMGB1 expression has been consistently linked with maintaining genomic integrity and radiation resistance in multiple cancers [7]. Mechanistically, HMGB1 appears to orchestrate DSB repair through multiple molecular axes that intersect with canonical repair pathways. In esophageal squamous cell carcinoma, HMGB1 itself is subject to oncogenic regulation, where the Wnt/β-catenin pathway is a key upstream activator. Upon ionizing radiation, β-catenin transactivates HMGB1 expression, which in turn promotes histone acetylation and enhances the DNA damage response to confer radioresistance [31]. Furthermore, HMGB1 was shown to promote radio-resistance by augmenting PARP1 levels that is closely linked to DNA single-strand breaks repair, while reduced PARP1 expression was documented upon HMGB1 knockdown accompanied by increased irradiation-induced DNA double-strand breaks [32].

Another recent evidence in SCLC demonstrates that HMGB1 promotes chemoresistance by driving PARP1 PARylation and nucleophagy (a selective autophagy type), thereby removing PARP1 from DNA lesions and allowing persistent repair evasion [33]. In breast cancer, HMGB1 depletion compromises DNA repair signaling by suppressing key kinases of the DNA damage response ATM and ATR, thereby attenuating strand break repair efficiency [34]. Similarly, HMGB1 silencing sensitized cells to ionizing radiation and resulted in increased γ-H2AX foci and longer comet assay tails, indicative of impaired DSB resolution in bladder cancers [35]. These findings suggest that HMGB1 enhances repair kinetics following radiation-induced damage, facilitating tumor cell survival under genotoxic stress. This role extends to chemotherapy, as evidenced in multiple myeloma, where HMGB1 knockdown increased DNA damage markers (γH2A.X) and decreased the key homologous recombination protein Rad51 as well as pro-survival autophagy following dexamethasone treatment [36]. These collective findings suggest that HMGB1 enhances repair kinetics following genotoxic insult, facilitating tumor cell survival under therapeutic stress.

Additionally, HMGB1’s contributions to DSB repair extend to the NHEJ pathway, whereby a key mechanism involves its direct association with the Ku70/Ku80 complex. In nasopharyngeal carcinoma, HMGB1 directly interacts with Ku70, a critical NHEJ factor, which promotes the DNA-binding activity of Ku70 and enhances the efficiency of the NHEJ repair pathway, contributing to resistance to ionizing radiation and cisplatin [37]. Accordingly, HMGB1 knockout or pharmacological inhibition with glycyrrhizin impairs NHEJ and re-sensitizes resistant nasopharyngeal cancer cells to both ionizing radiation and cisplatin [37]. This therapeutic effect is recently further demonstrated in colorectal cancer, where glycyrrhizin was shown to significantly downregulate the NHEJ pathway through direct inhibition of HMGB1, resulting in reduced cell viability and increased DNA damage [38]. Moreover, HMGB1’s role in facilitating NHEJ complex function is conserved in V(D)J recombination, where it directly associates with the Ku70/Ku80 complex, promoting the recruitment of a DNA-dependent protein kinase catalytic subunit (DNA-PKcs) to DSB sites that accelerates NHEJ and DNA repair [39]. This demonstrates that HMGB1, by enhancing both DSB repair capacity, enables tumor cells to survive and evade the cytotoxic effects of anti-cancer therapies.

### 2.2. HMGB1 in Nucleotide Excision Repair (NER)

HMGB1’s influence on DNA repair extends beyond DSB processing, functioning as a crucial cofactor in the NER pathway. It recognizes and binds cooperatively with the XPA–RPA complex at sites of drug-induced lesions such as 1,2-GG intrastrand cisplatin adducts and psoralen crosslinks (ICLs), and facilitates the recruitment of other core recognition factors, including XPC–RAD23B [40,41]. These interactions enhances the assembly of higher-order repair complexes and promotes error-free lesion processing, contributing to increased cell survival and reduced mutagenesis following genotoxic stress [13,42]. Consistent with these findings, HMGB1-deficient cells exhibit reduced NER efficiency, hypersensitivity to UVC and platinum agents, and persistent chromatin damage, highlighting its essential role in maintaining genomic integrity [42]. From a therapeutic perspective, this repair-enhancing activity may allow tumor cells to more efficiently remove platinum adducts, thereby promoting survival and chemoresistance.

However, HMGB1’s function in NER is context-dependent. Multiple studies have demonstrated that it can also bind tightly to cisplatin-DNA adducts and hinder their repair, particularly at early time points, thereby amplifying cisplatin cytotoxicity [43,44,45]. This apparent contradiction is governed by the cellular redox state, as the interaction between HMGB1 and cisplatin-damaged DNA depends on cysteine residues C23 and C45 within the A-box domain [46]. Under reducing conditions, HMGB1 binds strongly and shields DNA lesions from repair, while oxidation of these cysteines into a disulfide bond weakens binding, restoring access to repair factors [47,48]. Further complexity arises from PTMs that modulate HMGB1’s localization and DNA-binding capacity. Acetylation and phosphorylation promote HMGB1 release from chromatin and alter its interaction with cisplatin adducts [49,50], while O-GlcNAcylation at serine 100 enhances HMGB1 oligomerization on damaged DNA but paradoxically reduces accurate lesion repair, favoring error-prone processing [51]. Thus, intracellular redox potential and distinct modification states acts as a molecular switch determining whether HMGB1 sensitizes or protects cells from chemotherapy.

### 2.3. HMGB1 and Mismatch Repair (MMR)

The involvement of HMGB1 in the MMR pathway presents a complex picture, with evidence suggesting a facilitatory rather than essential role in this process. Biochemical reconstitution assays have shown that HMGB1 participates in the initial damage recognition phase by interacting with the core MMR proteins MLH1 and MSH2 [52]. HMGB1 also cooperates with replication protein A (RPA) to stimulate the exonuclease activity of EXO1, a critical step in mismatched DNA excision, and can even substitute for RPA in promoting EXO1-dependent repair [53].

Emerging evidence extends HMGB1’s role beyond canonical mismatch correction to the processing of helix-distorting DNA lesions such as interstrand crosslinks (ICLs). In these contexts, MMR proteins, particularly MSH2-MSH6 and MLH1-PMS2, function cooperatively with NER components like XPC-RAD23B and XPA-RPA to recognize and excise ICLs [54,55]. Hence, HMGB1 may stabilize MMR–NER interactions and enhance lesion recognition [52], acting as a structural facilitator rather than a core enzymatic factor. Critically, studies in HMGB1-deficient mouse embryonic fibroblasts indicate that MMR proceeds efficiently in its absence, suggesting that HMGB1 may be dispensable for core MMR activity [56]. Moreover, while HMGB1 capacity to stimulate EXO1 activity and bridge MMR-NER complexes suggests possible implications for therapy resistance, direct evidence linking HMGB1-mediated MMR modulation to therapy resistance remains lacking, warranting further mechanistic investigation in cancer-relevant systems.

### 2.4. HMGB1 in Base Excision Repair (BER)

Beyond its role in NER, HMGB1 modulates base excision repair BER through selective interactions with core repair enzymes. It binds to intermediates containing 5′-deoxyribose phosphate (5′-dRP) flaps and exhibits weak 5′-dRP lyase activity, yet inhibits single-nucleotide BER (SN-BER) by competing with DNA polymerase β for 5′-dRP binding [30,57]. In contrast, HMGB1 stimulates apurinic/apyrimidinic endonuclease 1 (APE1) and flap endonuclease 1 (FEN1) activities, thereby promoting long-patch BER (LP-BER) through enhanced strand incision and flap cleavage [30,58,59]. While such regulation influences genome stability and repeat expansion, there is currently no direct evidence linking HMGB1-mediated BER to therapy resistance in cancer, warranting further investigation.

Taken together, these findings on HMGB1 in DNA damage repair mechanisms delineate it as both a structural and signaling modulator of DNA repair. It acts at the intersection of chromatin remodeling, damage sensing, and repair complex recruitment, ensuring efficient resolution of strand breaks while buffering cells against genotoxic therapies. Consequently, elevated HMGB1 expression contributes to a radio-resistant phenotype, whereas its depletion renders cancer cells vulnerable to DNA-damaging treatments, suggesting that therapeutic targeting of HMGB1 may represent a potential promising frontier for sensitizing tumors to anti-cancer therapy [60]. Table 1 provides a summary of HMGB1 functions in DNA repair mechanisms and roles in therapy resistance.

## 3. HMGB1/Autophagy Interplay in Cancer Therapy Resistance: A Promising Target

Alongside its role in repairing therapy-induced DNA lesions, HMGB1 orchestrates a complementary cytoplasmic defense program that contributes to cancer therapy resistance through its capacity to stimulate autophagy under stress. Autophagy is a fundamental cellular process involving the dynamic rearrangement of membranes to form phagophores, autophagosomes, and ultimately autolysosomes, which degrade cellular components ranging from proteins and organelles to invasive pathogens [61]. In cancer, this process carries an inherent duality: it can suppress early tumor formation by preventing genomic and metabolic instability, yet in established tumors it often becomes a powerful survival mechanism [62,63,64]. Under conditions such as nutrient deprivation, oxidative injury, hypoxia, and exposure to DNA-damaging therapy, HMGB1-driven autophagy allows tumor cells to endure stress, evade apoptosis, and reconfigure their metabolism. Indeed, genetic deletion or silencing of HMGB1 in mouse embryonic fibroblasts and multiple cancer cell models leads to profound defects in autophagic flux, establishing its non-redundant role [6,29].

### 3.1. Nuclear HMGB1: Transcriptional Control of the Autophagic Machinery

HMGB1 regulates autophagy in a compartment-specific manner. Within the nucleus, nuclear HMGB1 directly influences the expression of key genes that govern the autophagic machinery and its selective forms. One of the best-characterized targets is HSPB1 (HSP27) since HMGB1 promotes its transcription, which in turn supports the cytoskeletal remodeling and vesicular dynamics required for autophagosome formation and trafficking [65]. This HMGB1–HSP27 axis plays a critical role in mitophagy that serves as a defense against metabolic stress, by the recruitment of Parkin to damaged mitochondria and enabling the ubiquitination of VDAC1 [65,66]. This process facilitates recognition of dysfunctional mitochondria by autophagy adaptors such as p62, which tether them to LC3-positive phagophores for selective clearance. Accordingly, loss of either HMGB1 or HSP27 produces similar phenotypes, characterized by fragmented mitochondria, reduced oxidative phosphorylation, and diminished ATP production, indicating impaired mitochondrial quality control and mitophagy [65,66].

Furthermore, evidence from human myeloid leukemia cells reveals that nuclear HMGB1-driven autophagy also involves transcriptional activation of genes encoding Beclin-1, VPS34, and UVRAG, which together form the PI3K III complex required for autophagosome nucleation. Mechanistically, HMGB1 enhances the transcriptional activity of stress-response genes by JNK and ERK, which ensures that autophagy can be rapidly mobilized to preserve mitochondrial stability and suppress apoptosis under drug-induced stress [67]. Accordingly, silencing HMGB1 reduces autophagic flux and enhances sensitivity to chemotherapeutic agents, highlighting its importance maintaining survival under drug-induced stress [67] (Figure 2).

### 3.2. Cytoplasmic HMGB1: Redox-Governed Modulation of Beclin-1 and Autophagy-Mediated Cancer Resistance

Following stress exposure, HMGB1 undergoes tightly regulated nuclear-cytoplasmic translocation that enables it to function as a direct effector of autophagy in the cytosol. This relocation is triggered by multiple upstream signaling events, including the accumulation of reactive oxygen species, activation of PARP1, and PTMs such as acetylation and poly-ADP-ribosylation. Chemotherapeutic agents, nutrient deprivation, and genotoxic stress all promote these modifications, loosening HMGB1’s interaction with chromatin and facilitating its export to the cytoplasm. The SIRT6-PARP1 axis is particularly important, where SIRT6-dependent stimulation of PARP1 poly-ADP-ribosylates HMGB1, enabling subsequent acetylation and efficient cytosolic translocation. Interference with this pathway prevents HMGB1 export and diminishes autophagy, thereby restoring sensitivity to chemotherapeutic drugs in leukemia [68]. Consistently, in cancer cells exposed to immuno-oncologic death ligands such as TRAIL, the PARP1-meidated HMGB1 cytosolic translocation axis maintains autophagic flux to prevent apoptosis, thereby contributing to TRAIL resistance. Conversely, pharmacologic inhibition of PARP1 or blockade of HMGB1 cytoplasmic translocation shifts the balance toward apoptotic cell death, strengthening therapeutic responses [69]. In support with these findings, Shen et al. demonstrated that blocking the HMGB1–PARP1–LC3 nucleophagy axis restores chemosensitivity and enhances responses to PARP inhibitors in SCLC [33], highlighting a therapeutic synergy worth exploring clinically (Figure 2).

Once in the cytosol, HMGB1 executes its pro-autophagic function primarily through a direct and regulated interaction with the core autophagy protein Beclin-1 [70,71]. This interaction is not merely associative but is functionally critical, as HMGB1 competes with the anti-apoptotic protein Bcl-2 for binding to Beclin-1. Therefore, by displacing Bcl-2, HMGB1 liberates Beclin-1 from its inhibitory complex, permitting the formation of the Beclin-1/VPS34 complex that is essential for the initiation of autophagosome formation [70,71]. This molecular switch from a Bcl-2-bound state to an HMGB1-bound state effectively direct the cellular machinery away from apoptosis and towards protective autophagy, enhancing cell survival under stress and resistance to therapies. For instance, the seminal work of Huang et al. in osteosarcoma demonstrated that HMGB1 binds Beclin-1 to promote ULK1/mAtg13/FIP200-dependent autophagy and chemoresistance [72].

Interestingly, the HMGB1/Beclin-1 interaction is exquisitely controlled by its redox state on specific cysteine residues. The C23-C45 intramolecular disulfide bond of HMGB1 maintains the protein in a conformation capable of productive Beclin-1 binding, as C23S and C45S mutants (which cannot form the disulfide bridge) abolish autophagy [70,71]. In contrast, the C106 residue controls subcellular localization rather than binding affinity, since C106S mutation biases HMGB1 toward the cytosolic compartment and leads to sustained autophagy, even in the absence of starvation cues [70]. Thus, HMGB1 containing a C23-C45 disulfide bridge and a reduced C106 residue serves as the optimal autophagy-activating conformation.

The process of HMGB1 translocation mediating Beclin-1 activation is also actively regulated by a network of upstream factors. For instance, the transcriptional regulator Yes-associated protein (YAP) promotes both the expression and the cytosolic translocation of HMGB1, thereby enhancing autophagy and tumor progression in gliomas [73]. Similarly, the protein phosphatase PPM1G has been identified as a direct upstream regulator that upregulates HMGB1 expression, thereby promoting both HMGB1-dependent autophagy and progression of pancreatic cancer [74]. Conversely, the telomeric protein TERF2/TRF2 impairs autophagy by binding to HMGB1 and sequestering it within the nucleus, thus preventing its cytosolic translocation [75]. Another layer of regulation is mediated by cellular kinases, as the release of Beclin-1 from Bcl-2 can be further reinforced by ERK- and JNK-dependent phosphorylation of Bcl-2, which reduces its affinity for Beclin-1 [76,77]. Moreover, at the post-transcriptional level, circular RNA G6PC3 was found to promote lung adenocarcinoma progression by physically binding to both HMGB1 and Beclin 1, thereby stabilizing their interaction and enhancing the formation of the pro-autophagic HMGB1/Beclin 1 complex [78]. Conversely, this signaling axis is negatively regulated by the tumor-suppressive microRNA MIR34A that directly targets the HMGB1 3 untranslated region to inhibit its expression, leading to suppressed autophagy and enhanced chemotherapy-induced apoptosis in retinoblastoma [79] (Figure 2).

Adding further complexity, the balance between HMGB1-mediated autophagy and apoptosis under stress is strongly shaped by its interaction with p53. HMGB1 and p53 can physically associate, with loss-of-function studies have illustrated their opposing regulatory roles. Mechanistically, p53 acts as a negative regulator of HMGB1-mediated autophagy, where its knockout increases cytosolic HMGB1 and enhances autophagic activity [80]. Conversely, HMGB1 can stimulate the nuclear accumulation of p53, since HMGB1 depletion elevates cytosolic p53 levels, suppressing autophagy and promoting apoptosis [81,82]. Thus, p53 acts as a negative regulator of HMGB1-mediated autophagy, while HMGB1 counters p53-dependent apoptotic signaling. The functional outcome is determined by whether HMGB1 remains nuclear or translocates to the cytoplasm.

Intriguingly, this dynamic can be subverted in the context of therapy resistance, as demonstrated by Luo et al. that HMGB1 mediates the shuttling of p53 to the cytoplasm for its subsequent autophagic degradation, which represents a key mechanism of sunitinib resistance [83]. The HMGB1–p53 interplay is also stress-intensity dependent, as in hepatoma cells treated with PCB quinone, low-dose stress favors nuclear HMGB1-driven autophagy, whereas high-dose stress shifts the response to p53-mediated apoptosis [84]. This indicates that HMGB1 and p53 are not fixed determinants of cell fate, but context-dependent modulators whose interaction fine-tunes survival versus death decisions, with initial autophagy activation predominantly acts as a pro-survival mechanism, while prolonged stress likely shifts the balance towards cell death (Figure 2).

Another context where HMGB1-mediated autophagy exhibits adaptive functions is under the oxidative stress, a common feature of tumor physiology, arising from oncogenic metabolism, hypoxia, and therapeutic intervenions. Under these conditions, cells rely on autophagy, with HMGB1 functions as a central redox-sensitive mediator in this critical stress response pathway. The research of Tang et al. demonstrated that hydrogen peroxide treatment or knockdown of the superoxide scavenger SOD1 promotes both cytosolic translocation and extracellular release of HMGB1, activating autophagic flux in mouse and human cell lines [85]. Accordingly, HMGB1 genetic loss or inhibition of its release significantly impairs the formation of autolysosomes and reduces autophagic activity, establishing HMGB1 as an essential component of the oxidative stress-autophagy axis [85].

In addition, the mechanistic basis for HMGB1’s role in oxidative stress-induced autophagy involves several pathways that implicate therapy resistance. For example, cytosolic HMGB1 mediates starvation-induced autophagy through the ROS/AMPK/mTOR signaling axis, regulating the degradation of the sodium/iodide symporter and contributing to radioiodine therapy resistance in thyroid cancer cells [86]. Similarly, in acute promyelocytic leukemia, ROS-induced HMGB1 translocation facilitates autophagic degradation of the oncogenic PML–RARα fusion protein through interactions with the ubiquitin-binding adaptor p62/SQSTM1, thereby enabling cell differentiation [87]. This regulatory network extends to long non-coding RNA mechanisms, where oxidative stress activates NORAD expression that enhances autophagy flux by stabilizing the ATG5-ATG12 complex through miR-433-3p sponging, contributing to oxaliplatin resistance in gastric cancer [88] (Figure 2). Collectively, cytosolic HMGB1 acts as a dynamic regulator of pro-survival autophagy in cancer cells exposed to anti-cancer therapies, which may represent a promising vulnerability to restore cancer sensitivity to treatment.

### 3.3. Extracellular HMGB1 as an Autocrine and Paracrine Mediator of Therapy Resistance

The role of HMGB1 in cancer therapy resistance extends beyond its intracellular functions to encompass critical signaling activities in the extracellular space. Upon its release from stressed, dying, or resistant cancer cells, HMGB1 acts as a prototypical DAMP molecule, primarily functioning to alert and activate the immune system by engaging pattern-recognition receptors [89,90]. However, in some tumor settings, this same receptor engagement can be co-opted by cancer cells and stromal components to activate inflammatory, pro-survival, and tissue-repair pathways that collectively enhance tumor adaptation to therapy [91]. Thus, in context-dependent manner, extracellular HMGB1 contributes to an adaptive microenvironment that can support tumor cell persistence under therapeutic stress.

The primary receptor mediating the effects of extracellular HMGB1 is RAGE, the binding to which initiates diverse downstream signaling cascades that converge on the induction of protective autophagy. A key pathway involves the activation of the MEK-ERK axis, as demonstrated by Huang et al. in colorectal cancer, where HMGB1 released from chemotherapy-damaged cells triggers RAGE-dependent ERK1/2 activation, which in turn phosphorylates the mitochondrial fission protein Drp1 at Ser616. This phosphorylation event promotes mitochondrial fission and facilitates autophagy, contributing to chemoresistance and tumor regrowth [92]. Studies in leukemia and Lewis lung carcinoma models showed that this RAGE-ERK signaling module is a conserved mechanism, where it sustains autophagic flux and promotes cell survival during nutrient stress and chemotherapy [93,94,95]. Furthermore, the HMGB1/RAGE axis not only activates ERK to inhibit mTOR and induce autophagy but also activates NF-κB to upregulate drug efflux pumps like P-glycoprotein (P-gp) and MRP, creating a multi-faceted defense system against chemotherapeutic agents in acute leukemia [95] (Figure 2).

Similar to intracellular HMGB1, the functional outcome of extracellular HMGB1-mediated autophagy is also critically determined by its redox state. The reduced form of HMGB1 binds to the RAGE and induces Beclin-1-dependent autophagy, promoting tumor cell resistance to diverse chemotherapeutic agents including alkylators, tubulin disruptors, and DNA intercalators [96,97]. By contrast, fully oxidized HMGB1 loses affinity for RAGE and Beclin1 and instead enhances drug cytotoxicity by promoting apoptosis through the caspase-9/-3 intrinsic pathway [96,97], positioning HMGB1 as a molecular switch that regulates autophagy and apoptosis within the tumor microenvironment.

Another pathway that can be engaged by the HMGB1/RAGE axis is PI3K/Akt, as it contributes to gemcitabine resistance in pancreatic cancer by promoting protective autophagy and upregulating the expression of the multidrug resistance protein MDR1. The therapeutic potential of blocking this specific pathway was demonstrated by the natural compound Lucidone, which was shown to inhibit the HMGB1/RAGE/PI3K/Akt axis, thereby suppressing both autophagic flux and MDR1 expression and consequently restoring chemosensitivity in gemcitabine-resistant pancreatic cancer cells [98]. This pathway’s broader significance is further highlighted in the context of physical ablation therapies. Following irreversible electroporation (IRE), a sublethal electric pulse can promote tumor recurrence by triggering HMGB1 release. In this setting, HMGB1 engages not only RAGE but also TLR4 to cooperatively suppress the PI3K/Akt/p70S6K axis and induce pro-survival autophagy. Critically, co-inhibition of both RAGE and TLR4 was required to disrupt this autophagic response, effectively sensitizing pancreatic cancer to IRE and underscoring the complexity and therapeutic relevance of extracellular HMGB1 signaling in treatment resistance [99]. The role of HMGB1 in modulating the Akt/mTOR pathway to foster therapy-induced protective autophagy is a conserved mechanism, as evidenced in synovial sarcoma, where HMGB1-mediated suppression of Akt/mTOR signaling was identified as the key mechanism decreasing sensitivity to oxymatrine, further validating this axis as a central target for intervention across diverse malignancies [100] (Figure 2).

In irradiated glioblastoma cells, released ATP activates the P2X7 receptor, which in turn promotes both DNA damage repair and the extracellular release of HMGB1. This establishes a P2X7-HMGB1-RAGE signaling axis that simultaneously enhances radioresistance through improved DNA repair and drives acquisition of malignant phenotypes like increased cell migration [101]. This is supported by the study of Li et al., which demonstrates that extracellular HMGB1 can promote more aggressive tumor phenotypes in glioblastoma. Mechanistically, HMGB1 induces the overexpression of p62 via a TLR4-p38-Nrf2 pathway. This p62 accumulation promotes the degradation of GSK-3β, leading to the stabilization of Snail and the induction of epithelial–mesenchymal transition (EMT), thereby linking HMGB1 signaling to increased invasion and metastatic potential [102].

The tumor microenvironment actively participates in this resistance mechanism. Cancer-associated fibroblasts (CAFs) in breast cancer have been shown to induce HMGB1 expression in cancer cells and contribute to the pool of extracellular HMGB1 following chemotherapy. This stromal-amplified HMGB1 secretion fosters doxorubicin resistance in breast cancer cells, an effect that can be reversed by HMGB1-neutralizing antibodies [103]. Furthermore, in ERα-positive breast cancer, tamoxifen itself can activate GPR30 on CAFs, triggering PI3K/AKT signaling that leads to the upregulation and secretion of HMGB1. This extracellular HMGB1 then activates MEK/ERK signaling in cancer cells, inducing pro-survival autophagy and conferring tamoxifen resistance [104]. Similarly, in hepatocellular carcinoma (HCC), tumor-derived HMGB1 engages TLR2 on macrophages, triggering a NOX2/ROS-dependent autophagic response that drives M2 macrophage polarization, thereby creating an immunosuppressive and pro-tumorigenic niche [105] (Figure 2).

### 3.4. The Autophagy-HMGB1 Secretion Loop: A Bidirectional Axis in Therapy Resistance

While HMGB1 is a well-established regulator of autophagy, a critical feedback loop exists wherein autophagy itself governs the secretion of HMGB1. This autophagic release of HMGB1 into the extracellular space can powerfully influence tumor cell survival, stromal communication, and ultimately, the therapeutic response, creating a self-reinforcing cycle of resistance.

A primary consequence of autophagy-mediated HMGB1 release is the direct enhancement of cancer cell survival. In gastric cancer, vincristine treatment induces autophagy, which subsequently promotes the extracellular release of HMGB1. This extracellular HMGB1 then acts through an autocrine or paracrine mechanism to transcriptionally upregulate the anti-apoptotic protein Mcl-1, thereby directly antagonizing chemotherapy-induced apoptosis [106]. This protective signaling is often mediated by HMGB1’s interaction with its receptor, RAGE, leading to the activation of downstream survival pathways such as ERK1/2, which further promotes tumor cell proliferation and chemoresistance [107]. The functional importance of this axis is underscored by studies showing that combined hyperthermia and chemotherapy in a hypoxic microenvironment promote autophagy and HMGB1 release in oral squamous cell carcinoma, and that inhibition of this pathway is a key component of effective combination therapy [108].

The tumor microenvironment amplifies this resistance loop, as shown in non-small cell lung cancer (NSCLC), where cancer-associated fibroblasts (CAFs) exhibit upregulated autophagy that is responsible for the secretion of HMGB1. This stromal-derived HMGB1 promotes the invasive and metastatic potential of cancer cells by activating the NF-κB signaling pathway [109]. Similarly, in luminal breast cancer, autophagic CAFs release HMGB1that activates TLR4 on cancer cells to maintain their stemness and tumorigenicity, linking stromal autophagy to the preservation of a treatment-resistant cancer cell population [110]. A particularly sophisticated mechanism of immune suppression driven by this axis involves the packaging of HMGB1 within autophagosomes under hypoxic conditions. In this context, tumor cell-released autophagosomes (TRAPs) are decorated with elevated levels of surface HMGB1. These HMGB1-carrying TRAPs then directly educate B cells to become IL-10-producing immunosuppressive cells, which in turn suppress the activity of both CD4^+^ and CD8^+^ T cells, creating a potent barrier to anti-tumor immunity and fostering an environment conducive to therapy resistance [111]. This creates a feed-forward loop, as the released HMGB1 can further stimulate autophagy in the CAFs themselves, sustaining the activated state of the tumor stroma [109].

However, the role of autophagy in HMGB1 release is also context-dependent and can, under specific conditions, lead to anti-tumor effects. In glioblastoma, temozolomide (TMZ) chemotherapy induces a process known as secretory autophagy, which selectively packages and releases HMGB1. Unlike the pro-tumorigenic release described above, this HMGB1 acts on tumor-associated macrophages (TAMs) via the RAGE-NF-κB-NLRP3 inflammasome pathway, driving them toward an M1-like, anti-tumor phenotype. This immunostimulatory environment enhances the chemosensitivity of glioblastoma cells to TMZ, positioning autophagy-mediated HMGB1 release as a favorable, therapy-promoting event [112]. This aligns with earlier foundational work of Thorburn et al., who demonstrated that autophagy can selectively regulate HMGB1 release in cells destined to die, and that this release is not a passive event but an active immunomodulatory signal. The secreted HMGB1 can enhance the immune response to otherwise poorly immunogenic apoptotic cells by activating a TLR4-dependent, tumor-specific immune response. This finding introduced the pivotal concept that therapeutic strategies could be designed to manipulate cell death mechanisms, specifically by promoting autophagy in dying cells, to ensure HMGB1 release, thereby stimulating an adaptive immune response that could seek out and destroy residual tumor cells, ultimately improving long-term treatment efficacy [113].

In conclusion, autophagy is a key driver of HMGB1 secretion, establishing a powerful bidirectional communication circuit within the tumor and its microenvironment. Typically, this loop promotes resistance by activating pro-survival signals in cancer cells and maintaining a supportive stromal niche. However, in specific therapeutic contexts, autophagic HMGB1 release can stimulate anti-tumor immunity and enhance treatment efficacy. This duality underscores that the functional outcome of the autophagy-HMGB1 axis is not fixed but is determined by the biological context, including the cell type, the nature of the stressor, and the state of the tumor immune microenvironment.

## 4. HMGB1 and Cell Death Mechanisms in Cancer Therapy Resistance

The HMGB1-driven programs of DNA repair and autophagy ultimately serve to delay or prevent the activation of regulated cell death pathways. However, when these protective mechanisms are overwhelmed, the cell’s fate is determined by the intricate interplay between HMGB1 and the core machinery of cell death. Here, HMGB1 again functions as a central switch, modulating sensitivity to apoptosis, necroptosis, ferroptosis, and pyroptosis.

### 4.1. HMGB1 in Apoptosis and Therapy Resistance

Apoptosis is a central barrier to malignant progression, yet cancer cells frequently circumvent it through adaptive stress-response pathways. HMGB1, once viewed as a nuclear structural protein, has emerged as a dynamic regulator of apoptotic sensitivity whose actions, once again, depend on its subcellular localization, PTMs, and redox status [29,97,114]. Across multiple tumor types, HMGB1 modulates both intrinsic and extrinsic apoptotic pathways, thereby promoting resistance to chemotherapy, radiotherapy, and targeted agents.

A major anti-apoptotic function of HMGB1 is its regulation of the intrinsic mitochondrial pathway. The integrity of the mitochondrial outer membrane is governed by the balance of Bcl-2 family proteins, and HMGB1 acts directly upstream of this checkpoint. In vemurafenib-resistant thyroid cancer cells, elevated HMGB1 suppresses caspase-3 activation and maintains mitochondrial function, reducing drug sensitivity [115]. HMGB1 frequently enforces a pro-survival transcriptional profile by upregulating Bcl-2 and suppressing Bax, which contributes to radioresistance in esophageal carcinoma and correlates with poor prognosis [116,117]. Similar mechanisms operate in hematological malignancies, where HMGB1 overexpression increases Bcl-2 levels and inhibits caspase-3/9 activation during adriamycin treatment, while HMGB1 knockdown restores chemosensitivity in resistant leukemia cells [118,119]. In gastric cancer, chemotherapy-induced HMGB1 release elevates Mcl-1 transcript levels, preserving mitochondrial integrity and preventing vincristine-induced apoptosis [106]. Through these coordinated actions on Bcl-2 family proteins, HMGB1 stabilizes mitochondrial membranes and restrains cytochrome c–caspase-9 signaling [28,65] (Figure 3).

HMGB1 also modulates the extrinsic apoptotic pathway. It can protect cells from death receptor signaling induced by CD95 and TRAIL, suppressing downstream activation of caspase-8 and Bax [120]. During TRAIL-induced apoptosis, PARP1-mediated poly–ADP-ribosylation drives HMGB1 translocation to the cytoplasm, where it binds Beclin-1 and diverts signaling toward autophagy rather than caspase-8 activation. Inhibition of this PARP1–HMGB1–Beclin-1 axis restores TRAIL-induced apoptosis in vitro and in vivo [69]. More broadly, HMGB1 promotes mitophagy to eliminate damaged mitochondria before apoptotic signals can propagate [66,121]; the PARP1/HMGB1 node acts as a molecular switch that biases cells toward autophagy over apoptosis and contributes to chemoresistance in hepatocellular carcinoma [69,122,123]. HMGB1-driven JNK and ERK activation in leukemia and colorectal cancer reinforces this autophagy-dependent apoptotic escape [67,124], while an HMGB1/c-Myc circuit mediates paclitaxel resistance in castration-resistant prostate cancer [125] (Figure 3).

The interplay between HMGB1 and p53 constitutes another decisive determinant of apoptotic competence. HMGB1 binds p53 under stress conditions, restricting its pro-apoptotic transcriptional activity and favoring autophagy [80,82]. HMGB1 can also promote p53 degradation through autophagy, enabling resistance to kinase inhibitors such as sunitinib [83]. Conversely, loss of p53 enhances cytosolic HMGB1 translocation and its binding to Beclin-1, further amplifying survival autophagy. HMGB1 inhibition reverses this cycle, leading to cytosolic p53 accumulation and restored apoptosis [81,82]. 

In addition to regulating the balance between apoptosis and autophagy, emerging evidence indicates that HMGB1/p53 crosstalk also intersects with cellular senescence, a stress-induced growth arrest program that prevents the proliferation of damaged cells [126]. Senescence is increasingly understood not as a strictly irreversible arrest but as a dynamic state characterized by persistent DNA-damage signaling, chromatin remodeling, resistance to apoptosis, and the development of a SASP, providing an alternative survival outcome in tumor cells subjected to genotoxic therapies [127]. Notably, senescent cells exhibit p53-dependent nuclear-to-cytosolic relocalization and secretion of HMGB1, which contributes to SASP induction through TLR4–NF-κB–mediated cytokine release [128]. Under DNA-damaging conditions, HMGB1 can tilt the cell-fate decision toward senescence by promoting p21-dependent arrest, while HMGB1 depletion shifts cells toward apoptosis [129]. This can be mediated by enhancing a STING–STAT6–p21 signaling axis that enforces therapy-induced senescence [130]. Together, these findings suggest that the HMGB1–p53 axis not only shapes apoptosis–autophagy transitions but may also influence the propensity of cancer cells to enter a senescent, therapy-tolerant state, thereby adding another dimension to HMGB1-mediated stress adaptation and resistance.

HMGB1’s redox state further dictates apoptotic outcomes. Reduced HMGB1 binds Beclin-1 and activates autophagy, or, when extracellular, engages RAGE to trigger PI3K and ERK1/2 survival pathways [70,92,97,131]. This promotes resistance across melanoma, pancreatic, and colon cancers [96,99,131,132]. Oxidation of HMGB1, especially at Cys106, disrupts Beclin-1 binding and instead promotes mitochondrial apoptosis, enhancing chemosensitivity [84,96,132,133]. Redox state also shapes immune-mediated apoptotic responses: reduced HMGB1 acts as an immunogenic DAMP via TLR4 to promote dendritic cell activation, whereas oxidized HMGB1 becomes non-immunogenic and supports immune evasion [133,134]. However, this extracellular signaling can also establish a chemoresistant niche, as demonstrated in prostate cancer, where HMGB1 released from docetaxel-treated dying cells engages TLR4/RAGE on surviving tumor cells to activate NF-κB and induce secretory clusterin (sCLU), which in turn sequesters Bax and potently inhibits apoptosis [135].

Tumor-microenvironment interactions further reinforce HMGB1-mediated apoptosis resistance. A gankyrin/NONO/AR/HMGB1/IL-6/STAT3 loop in prostate cancer maintains HMGB1 expression and TAM-driven survival signaling [136]. In breast cancer, macrophage-derived CXCL1 activates an IGF1R/STAT3/HMGB1 axis that promotes autophagy-mediated paclitaxel resistance and inhibits apoptosis [137]. In lung adenocarcinoma, cisplatin-persistent cells accumulate nuclear HMGB1 and upregulate RAGE, indicating a coordination between extracellular immunomodulation and intracellular DNA damage repair signaling [138]. Additionally, radiation-induced HMGB1 secretion activates CAFs via TLR4/PI3K-AKT to suppress apoptosis and drive cancer radioresistance, an effect reversed by HMGB1 inhibition [139] (Figure 3).

Non-coding RNA networks represent another layer that fine-tunes HMGB1 levels and thereby controls apoptotic sensitivity. Multiple microRNAs, miR-142-3p, miR-34a, miR-129-5p, and miR-451, directly target HMGB1 to restore caspase activation and suppress autophagy in doxorubicin-resistant breast cancer, AML, retinoblastoma, and pediatric AML [79,140,141,142,143,144] (Figure 3). Conversely, oncogenic lncRNAs such as ANRIL and HOTAIR derepress HMGB1 by sponging these microRNAs, promoting autophagy-dependent chemoresistance [145,146]. In multiple myeloma, MALAT1 protects HMGB1 from ubiquitin-mediated degradation, elevating HMGB1 levels to drive autophagy and suppress apoptosis, contributing to resistance to chemotherapeutic agents [147]. HMGB1 mRNA itself can function as a ceRNA for the miR-200 family, derepressing RICTOR and attenuating apoptosis while impairing anti–PD-L1 therapy through PD-L1+ exosome production [148].

Finally, therapeutic inhibition of HMGB1 highlights its integral role in maintaining apoptosis resistance. Glycyrrhizin, a direct HMGB1 inhibitor, suppresses NHEJ repair, induces DNA damage, restores caspase-3 activation, and triggers apoptosis in colorectal cancer models [38], illustrating the therapeutic value of targeting the HMGB1-driven survival network. Altogether, HMGB1 is a central integrator of stress signals that governs the balance between apoptosis and pro-survival mechanisms that enable cancer cells to resist therapeutic-induced death. Disrupting these specific, pro-tumorigenic functions of HMGB1 may represents a promising strategic avenue for overcoming therapy resistance.

### 4.2. HMGB1 and Necrosis/Necroptosis in Cancer Therapy Resistance

Whereas necrosis represents an accidental, unregulated form of cell death, necroptosis is a genetically programmed pathway that can be activated when apoptosis is blocked. Whether cells are necrotic or necroptotic, they release HMGB1 as a DAMP in the extracellular milieu, triggering complex signaling through receptors like RAGE and TLR4 that can either activate immune response or paradoxically enhance tumor survival and foster therapy resistance [114,149,150]. The fundamental distinction in HMGB1 release between necrotic and apoptotic cell death establishes its complex role in necroptosis-associated inflammation. While necrotic cells passively release HMGB1 as a potent inflammatory signal, apoptotic cells retain HMGB1 bound to chromatin, preventing inflammation [8,114]. This mechanistic difference also underscores HMGB1’s unique position as a key mediator of the immune response to necroptotic cell death. However, the consequences of this HMGB1 release can drive pro-tumor effects, as for example, HMGB1 released from necrotic keratinocytes triggers TLR4-dependent inflammation that is associated with enhanced tumor development, with TLR4 expression required on both immune and radio-resistant cells for full carcinogenesis [151]. Furthermore, upstream regulators of necrosis can critically influence HMGB1 release. For instance, metabolic stress induced by the deficiency in the mitochondrial deacetylase sirtuin 3 (SIRT3) enhances necroptosis, and activates the NLRP3 inflammasome, a process that promotes HMGB1 release and amplifies inflammatory signaling, thereby linking mitochondrial dysfunction directly to the necrotic release of this key DAMP and its therapy resistance-promoting roles [152,153,154].

Importantly, while necrotic death augments HMGB1 release, extracellular HMGB1 can function as a survival signal that suppresses and prevents further necroptosis. This was shown by Liu et al. in acute myeloid leukemia (AML), where etoposide-induced necroptosis promotes HMGB1 release, yet the extracellular HMGB1 itself activates NF-κB signaling that prevents necroptosis in surrounding AML cells, creating a self-limiting death process that may contribute to therapeutic resistance [155]. In addition, within the tumor microenvironment, hypoxia further amplifies the necroptosis-HMGB1 axis to drive resistance and metastasis. Hypoxic conditions promote HMGB1 release from necrotic tumor cells [156], which in turn recruits M2-like macrophages and creates an IL-10 rich milieu through RAGE signaling, ultimately facilitating tumor growth and metastasis [157]. The persistent activation of RAGE by HMGB1 released from necrotic tumor cells further reinforces resistance by inducing pro-survival autophagy and inhibiting apoptotic pathways [158,159], creating a feedback loop that maintains the resistant phenotype. Hence, HMGB1-necroptosis axis critically impact the therapeutic outcomes, where the initial inflammatory signal triggered by HMGB1 release can be subverted to support tumor survival, immune evasion, and treatment resistance across several cancer types (Figure 3).

### 4.3. HMGB1 in Ferroptosis and Cancer Therapy Resistance

Beyond apoptosis, resistance to therapy can also emerge through suppression of alternative cell death pathways [160,161]. Ferroptosis, an iron-dependent, lipid peroxidation-driven form of regulated necrosis defined by Fenton reaction-mediated ROS generation, polyunsaturated lipid peroxidation, and glutathione peroxidase 4 (GPX4) suppression [162,163]. In cancer, ferroptosis inhibition is directly associated with therapy resistance [164,165,166], and HMGB1 has emerged as an integrator of ferroptotic signaling, inflammatory activation, and therapy resistance. Bioinformatics analyses across cancer types, including colorectal cancer, consistently identify HMGB1 as a key ferroptosis-related gene, with its elevated expression in tumors correlating with a ferroptosis-resistant phenotype and poor prognosis, underscoring its clinical relevance as a resistance factor [167,168,169].

Within the cell, HMGB1 regulates core ferroptotic determinants by modulating GPX4-centered antioxidant defenses and influencing ACSL4-dependent lipid peroxidation. A recent study by Zhao et al. demonstrated that HMGB1 function as a ferroptosis suppressor in hepatocellular carcinoma, while Astragaloside IV directly binds HMGB1, represses its expression, and thereby induces GPX4-dependent ferroptotic death in hepatocellular carcinoma, suggesting that HMGB1 can act as a protective brake against oxidative collapse [170]. Yet, a body of evidence supports a ferroptosis-promoting role for HMGB1. For instance, BMAL1 suppresses HMGB1 to maintain GPX4 activity and chemoresistance in AML, whereas BMAL1 knockdown upregulates HMGB1 and restores ferroptotic sensitivity [171]. Another study showed that SIRT1 prevents HMGB1 cytosolic translocation and elevates ACSL4, thereby blocking lipid peroxidation and mediating cytarabine resistance, while SIRT1 inhibition reinstates HMGB1-dependent ferroptosis and therapeutic response [172]. Additional regulators such as pirin (PIR) exert similar control, as PIR depletion enhances HMGB1 cytosolic translocation and activates Beclin1-ACSL4 autophagic machinery to potentiate ferroptosis in pancreatic cancer [173].

Beyond intracellular functions, HMGB1 is secreted as a result of ferroptotic cell death as a DAMP, acting as a ferroptosis marker. Ferroptosis inducers, including erastin, sorafenib, RSL3, and FIN56, stimulate HMGB1 release through an autophagy-dependent mechanism in which ATG5/ATG7 activity and autophagy-mediated HDAC inhibition drive HMGB1 acetylation and secretion [174]. The extracellular functions of HMGB1 exhibit dual consequences that critically determine the therapeutic outcome and immune microenvironment activities, driving both anti-tumor and pro-tumor immune response, depending on the context. For example on HMGB1 roles in immunosuppression, studies delineating the temporal stages of ferroptosis reveal that HMGB1 release is a hallmark of the ‘terminal’ phase, and that cells undergoing ferroptosis can potently suppress dendritic cell (DC) maturation, antigen cross-presentation, and subsequent anti-tumor T-cell immunity, thereby actively impeding the adaptive immune response [175]. HMGB1 can further exacerbate immunosuppression by driving their conversion to a CD11c^low^ CD45RN^high^ tolerogenic state, repressing T-cell activation [176]. Ferroptotic HMGB1 also promotes immunosuppressive remodeling by recruiting myeloid-derived suppressor cells (MDSCs) and upregulating PD-L1 expression, fostering immune escape in hepatocellular carcinoma [177]. 

Furthermore, in pancreatic cancer, ferroptosis-driven release of oncogenic KRAS^G12D^ protein, a process facilitated by HMGB1-associated autophagic secretion, educates tumor-associated macrophages towards a pro-tumor M2-like phenotype via a RAGE-STAT3 signaling axis, creating a resistant, tumor-promoting niche [178]. In support with this findings, ferroptosis inducers activate a prognostically adverse gene signature that includes HMGB1, linking its release to pathways of extracellular matrix remodeling and therapy resistance in aggressive breast cancer subtypes [179,180]. This collective evidence has motivated rational combination strategies involving HMGB1 blocking to overcome immunotherapy resistance. A recent nanoplatform designed by Sha et al. exemplifies this approach by simultaneously inducing tumor-selective ferroptosis through Fe^2+^/ROS generation and blocking HMGB1 release via glycyrrhizic acid loading, thereby attenuating ferroptosis-driven MDSCs recruitment and synergistically boosting immune checkpoint blockade efficacy [181] (Figure 3). Accordingly, these studies suggest that HMGB1 inhibition may be a promising approach to overcome key mechanisms of resistance to chemo- and immunotherapies in specific cancer contexts.

Conversely, HMGB1 release during ferroptosis can promote immunogenic cell death (ICD) by enhancing dendritic-cell maturation, promoting antigen presentation, and priming CD8^+^ T-cell responses, thereby amplifying the antitumor effects of ferroptosis-inducing therapies such as auranofin and GPX4-targeted degraders [182,183,184]. Accordingly, newly emerging studies reinforce this immunostimulatory potential, such as in pancreatic ductal adenocarcinoma, where the inhibition of macrophage-capping protein (MCP) suppresses PIR, downregulates GPX4, and drives robust ferroptosis accompanied by accelerated HMGB1 cytosolic accumulation and release. Released HMGB1 then uniquely activates pro-inflammatory M1-like macrophage polarization, via binding RAGE rather than TLR4 [174], enabling ferroptosis to function as both a direct tumoricidal mechanism and an immunogenic trigger [185] (Figure 3).

Collectively, these findings position HMGB1 as a context-sensitive regulator of ferroptosis, which implicates cancer resistance to various therapies. HMGB1 intracellular functions can modulate GPX4 activity and lipid peroxidation, while its extracellular release dictates whether ferroptotic death culminates in immunogenic tumor clearance or in immunosuppressive escape. This complex duality explains why HMGB1 can both potentiate ferroptosis-induced tumor clearance and, under different conditions, maintain resistance by stabilizing antioxidant defenses or shaping an immune-evasive microenvironment. As HMGB1-ferroptosis axis continues to expand mechanistically, further investigations are required to resolve the determinants and identify biomarkers of its pro- and anti-ferroptotic functions, as well as determine specific HMGB1 modulation to overcome therapy resistance in refractory cancers.

### 4.4. HMGB1 in Pyroptosis and Cancer Therapy Resistance

The relationship between HMGB1 and pyroptosis in cancer therapy presents a complex paradox, wherein this DAMP can function as either an antagonist or protagonist of treatment efficacy, largely determined by contextual factors within the tumor microenvironment. Under specific conditions, HMGB1 release during pyroptosis can paradoxically fuel tumor progression and confer therapy resistance. In colitis-associated colorectal cancer, GSDME-mediated pyroptosis promotes tumorigenesis through the release of HMGB1, which subsequently activates the ERK1/2 pathway to drive tumor cell proliferation [186]. Similarly, in neuroblastoma, endogenous HMGB1 regulates chemotherapy-induced pyroptosis through the ROS/ERK1/2/caspase-3/GSDME signaling axis, where, surprisingly, its knockdown inhibited chemotherapy-induced pyroptosis and switched the cell death mode to apoptosis, and thereby increased sensitivity to chemotherapeutic agents [187]. This suggests that in certain contexts, HMGB1-mediated pyroptosis may represent an adaptive escape mechanism from more lethal forms of cell death like apoptosis. The pro-tumorigenic potential of pyroptosis-derived HMGB1 extends to microenvironment remodeling, as evidenced in the recent study of Gao et al. in colorectal cancer, where Caspase-6/GSDMC-mediated tumor cell pyroptosis leads to HMGB1 release that enhances CXCL2 expression, subsequently recruiting MDSCs to create an immunosuppressive microenvironment conducive to tumor progression and immune evasion [188].

Conversely, substantial evidence demonstrates that HMGB1 release during pyroptosis can potentiate anti-tumor immunity and treatment response. In BRAF-mutant melanoma, combinatorial targeted therapy induces GSDME-mediated pyroptosis with HMGB1 release, creating an immune-stimulatory microenvironment characterized by enhanced T cell and activated dendritic cell infiltration [189]. The immunogenic potential of this pathway is further highlighted by studies showing that environmental toxins like PCB29-pQ induce pyroptosis through HMGB1-TLR4-NLRP3-GSDMD signaling [190], while in leukemia, arsenic trioxide-induced pyroptosis promotes HMGB1 release that activates natural killer cell-mediated immunogenic cell death [191]. The therapeutic relevance of this immunogenic axis is particularly evident in breast cancer, where neoadjuvant chemotherapy agents induce GSDME-mediated pyroptosis accompanied by HMGB1 release, resulting in enhanced phagocytosis by macrophages and increased secretion of IFN-γ and IL-2 [192].

This duality in HMGB1 function during pyroptosis, whether promoting resistance through proliferative signaling and immunosuppression or enhancing sensitivity through immunogenic cell death, appears to depend on multiple factors including cancer type, genetic background, therapeutic agent, and the tumor microenvironment. Currently, research into the HMGB1-pyroptosis axis in the context of established drug-resistant cancers remains in its infancy. Therefore, future investigations are required to define the precise molecular and contextual rules governing intracellular HMGB1 interplay with pyroptosis and whether pyroptotic HMGB1 release fosters a resistant niche or a therapeutic breakthrough. A deeper understanding of this interplay will be essential to inform strategies that either suppress its pro-tumorigenic signals or harness its immunogenic potential to overcome therapy resistance.

## 5. HMGB1 as a Molecular Driver of Multidrug Resistance Proteins (MRPs)

While modulating DNA repair, autophagy, and cell death pathways allows cancer cells to survive the cytotoxic effects of chemotherapeutics, a parallel resistance strategy involves simply reducing intracellular drug concentrations. This is achieved through the upregulation of MDR proteins, a process that is itself powerfully influenced by the inflammatory and stress-signaling pathways that can be regulated by HMGB1. MDR represents one of the most formidable obstacles in cancer therapy, largely due to the overexpression of multidrug resistance proteins (MRPs), such as ATP-binding cassette (ABC) transporters, particularly P-glycoprotein (P-gp/ABCB1), members of the MRPs family (MRPS1–9), and ABCG2, that actively efflux chemotherapeutic drugs [193]. The induction of these MRPs is strongly influenced by chronic inflammation, which is now recognized as a key determinant of tumor cell survival, immune evasion, and treatment failure [2,194]. 

At the center of this inflammatory-resistance interface lies the transcription factor NF-κB that is typically activated by inflammatory cytokines, prostaglandins, and stress mediators present in the tumor microenvironment, which in turn promotes the expression and activity of key MRPs [4,195]. NF-κB directly binds the MDR1/ABCB1 promoter and enhances its transcription, and multiple studies confirm that NF-κB activation is tightly coupled to increased P-gp expression and drug efflux [195,196,197,198]. Inhibition of NF-κB consistently suppresses P-gp expression and restores chemosensitivity, as shown in hepatocellular carcinoma where metformin or NF-κB siRNA downregulate MDR1 and resensitize resistant cells [199]. Natural products such as γ-tocotrienol, ferulic acid, and morellic acid B likewise reverse MDR by inhibiting NF-κB activation and thereby suppressing P-gp transcription and efflux function [200,201,202], establishing inflammation-driven NF-κB signaling as a dominant pathway controlling the transcriptional and functional upregulation of MRPs.

In this regard, a central upstream activator of NF-κB is HMGB1, the multifunctional DAMP whose release profoundly reshape the drug response landscape of tumors [125,203,204]. Numerous studies demonstrate that HMGB1, upon release from stressed or dying tumor cells, engages RAGE, TLR4, or TNFR1 to initiate canonical NF-κB activation, resulting in phosphorylation of IKKα/β, degradation of IκBα, and nuclear translocation of p65 [92,204,205]. In gastric cancer, HMGB1 knockdown reduces NF-κB activity and suppresses proliferation and invasion, confirming that NF-κB operates downstream of HMGB1 signaling [206]. Similarly, HMGB1 binding to TNFR1 activates NF-κB to drive tumor progression and castration resistance in prostate cancer [205]. Likewise, HMGB1/TLR4/NF-κB signaling mediates metastasis and enhances inflammatory cytokine production in breast cancer [204], while drug-resistant myeloma cells exhibit increased HMGB1 release, whereby silencing HMGB1 diminishes NF-κB phosphorylation and restores drug sensitivity [207]. Together, these findings establish HMGB1 as a central regulator of NF-κB activation in cancer, proposing a link between cellular stress and inflammation to MRPs expression-mediating therapy resistance through a convergent HMGB1–NF-κB axis.

Indeed, multiple lines of evidence indicate that extracellular HMGB1 directly induces MDR through RAGE-mediated activation of the canonical NF-κB pathway. Binding of HMGB1 to RAGE leads to IKKα/β phosphorylation, IκBα degradation, and nuclear translocation of p65/p50, which in turn initiates transcription of ABCB1/P-gp and multiple MRPs family members [95]. In acute leukemia, HMGB1/RAGE signaling increases P-gp and MRPs expression, concurrently enhancing autophagy and suppressing apoptosis to create a survival-permissive state. Accordingly, blocking HMGB1 or its receptors reduces NF-κB phosphorylation and substantially decreases efflux transporter expression [95]. Furthermore, HMGB1 overexpression in NSCLC correlates with increased P-gp expression, enhanced proliferation, and cisplatin resistance, further linking cytoplasmic HMGB1 to multidrug-resistance phenotypes [208]. Intriguingly, another resistance mechanism to paclitaxel is mediated by sustained HMGB1 expression and release that activates c-Myc signaling, which is a direct transcriptional driver of ABC transporters and metabolic pathways essential for drug efflux and proliferation [125,209,210]. 

Beyond NF-κB and c-Myc activation, recent studies demonstrated that HMGB1 drives a second major MDR-promoting program through the induction of cytoprotective autophagy by intracellular HMGB1. In colorectal cancer, HMGB1-dependent autophagy markedly upregulates ABCG2, P-gp, MRP2, and GST-π, whereas autophagy inhibition by 3-MA reverses these effects and restores drug sensitivity [211]. Mechanistically, tumor cells suppress the transcriptional repressor Gfi-1, which normally inhibits HMGB1 expression, thereby amplifying HMGB1-driven autophagy and concurrently elevating multiple efflux transporters [211].

In addition, tumor-associated macrophage-derived CXCL1 adds a further layer of regulation by activating IGF1R, leading to STAT3 phosphorylation and subsequent transcriptional upregulation of HMGB1, which drives autophagy and increases ABCG2 expression. In this study, Yang and colleagues also demonstrated that silencing CXCL1 or blocking IGF1R/STAT3 signaling disrupts this HMGB1–autophagy–MRPs axis and restores drug sensitivity in breast cancer [137]. Moreover, although not in the cancer context, HMGB1 was shown drives upregulation of P-gp at the blood–brain barrier following cerebral ischemia via the related TLR4/NF-κB pathway [212]. This supports that HMGB1 functions as a potent upstream regulator of a well-established NF-κB–MRPs-therapy resistance axis, further solidifying HMGB1’s position as a central and versatile driver of therapy molecular resistance in cancer.

## 6. Therapeutic Strategies Targeting HMGB1 to Overcome Therapy Resistance

Because HMGB1 occupies nodal positions at the interplay of stress sensing, inflammation, autophagy and DNA-damage responses, it can serve as an attractive target for strategies aiming to re-sensitize tumors to chemotherapy, radiotherapy and immunotherapy. The multifaceted resistance program orchestrated by HMGB1, as detailed in the previous sections, provides a strong rationale for its therapeutic inhibition and reveals a wide variety of potential therapeutic interventions. Specifically, peptides, small molecules, repurposed drugs, natural products and nanoplatforms, have been developed to block HMGB1 expression, prevent its translocation and release, neutralize extracellular HMGB1, or interfere with HMGB1-receptor signaling. These approaches work primarily by either (i) blunt HMGB1-driven NF-κB/ERK/STAT3 survival signaling and thereby reduce expression of pro-survival genes and MDR transporters, (ii) inhibit HMGB1-dependent autophagy that sustains drug tolerance, or (iii) rewire HMGB1-mediated immune signals to favor immunogenic cell death (ICD) rather than immune suppression [20,29,213].

Targeting the HMGB1 pathway with its endogenous A-box peptide has emerged as a promising strategy to combat therapy resistance, with mechanisms that extend beyond simple receptor blockade. Initially characterized as a competitive antagonist, the Box A peptide functions by occupying HMGB1 receptors such as RAGE on the cell surface, thereby preventing the pro-tumorigenic signaling of the full-length protein. This extracellular mechanism has demonstrated significant anti-tumor and anti-angiogenic efficacy, notably improving survival in preclinical models like mesothelioma [214,215]. Subsequent research has revealed a more complex, dual mode of action. Beyond its extracellular role, intracellular Box A expression exerts a direct genotoxic effect specifically in cancer cells, inducing DNA double-strand breaks marked by γH2AX foci and activating the ATM/ATR DNA damage response, which culminates in apoptosis [216]. This paradigm was further advanced by a recent gene therapy approach in cisplatin-resistant non-small cell lung cancer (NSCLC). In this context, Box A expression not only induced DNA damage but also critically altered the subcellular localization of endogenous HMGB1, reducing its pro-survival cytoplasmic presence and thereby re-sensitizing resistant cells to cisplatin-induced death [217]. Consistent with these findings, cisplatin-sensitive A549 cells undergo dose-dependent cytoplasmic translocation of HMGB1, whereas resistant A549/DDP cells maintain high cytoplasmic HMGB1 levels; HMGB1 knockdown restored sensitivity to cisplatin, 5-FU, and oxaliplatin [218]. These data suggest Box A peptide as a multifaceted promising therapeutic agent capable of disrupting HMGB1-mediated therapy, encouraging its use in combination therapies.

An alternative strategy to mitigate the pathological effects of HMGB1 involves the use of inhibitory peptides that directly sequester the protein and suppress its pro-inflammatory signaling. For instance, the HBHP peptide has demonstrated efficacy across pathological contexts, where it was initially shown to inhibit microglial cell activation and the subsequent release of pro-inflammatory cytokines, highlighting its ability to dampen neuroinflammation by targeting HMGB1 [219]. Our own work has further expanded its therapeutic profile, demonstrating that HBHP disrupts the HMGB1-Hsp70 DAMP complex in lung and colorectal cancers [203]. This disruption effectively inhibits downstream prostaglandin E2 signaling-mediated inflammation and pro-survival autophagy in residual tumor cells, ultimately suppressing their repopulation capacity and reducing tumor recurrence in vitro and in vivo [203]. Another example is the Ac2-26 peptide, which has shown protective efficacy against hepatic ischemia–reperfusion injury by directly binding to HMGB1 and attenuating its activation of the pro-inflammatory TLR4/NF-κB signaling axis [220]. Notably, its function extends beyond immunomodulation, as recent evidence reveals a direct anti-tumor role in cervical carcinoma. In this context, Ac2-26 inhibits cancer cell proliferation and drives a distinct gene expression profile, most significantly through the downregulation of the pro-oncogenic factor ID1 and its pathway component BMPR1B [221].

Small molecules and natural compounds that directly target HMGB1 represent a promising therapeutic strategy for overcoming cancer therapy resistance. The natural compound glycyrrhizin (GL) and its derivatives function as potent HMGB1 inhibitors by binding to the protein and disrupting its pathological interactions. This mechanism has been validated across diverse resistant cancers. In castration-resistant prostate cancer, GL counteracts paclitaxel resistance by blocking the release of HMGB1, thereby disrupting the sustained activation of the downstream c-Myc oncogenic pathway [125]. Its efficacy extends to DNA damage response pathways, as GL sensitizes colorectal cancer and nasopharyngeal carcinoma (NPC) to treatment by suppressing the NHEJ repair mechanism. In NPC, this occurs via GL’s disruption of the critical HMGB1-Ku70 interaction, while in colorectal cancer, it inhibits HMGB1 to promote DNA fragmentation and apoptosis [37,38]. Beyond direct HMGB1 binding, GL also exhibits a unique, HMGB1-independent function by directly binding to progesterone receptor membrane component 1 (PGRMC1), a heme-binding protein implicated in chemoresistance, thereby inhibiting PGRMC1-mediated EGFR activation [222]. Furthermore, GL disrupts the tumor microenvironment by inhibiting the autophagy-dependent release of HMGB1 from cancer-associated fibroblasts (CAFs), which in turn suppresses NF-κB-driven invasion and metastasis in non-small cell lung cancer models [109]. In colorectal carcinogenesis, GL treatment attenuates HMGB1-TLR4-NF-κB signaling, leading to reduced inflammation, DNA damage, and cancer stem cell proliferation [223].

The small molecule ethyl pyruvate (EP) exerts broad anti-tumor effects primarily by inhibiting the nucleocytoplasmic translocation and release of HMGB1. This action disrupts key resistance pathways across diverse cancers, as EP reverses cisplatin resistance in cervical and lung cancer models, switching cell death from a pro-inflammatory necrotic phenotype to apoptosis and disrupting pro-survival autophagy [224,225]. Furthermore, EP overcomes prostate cancer resistance to castration by suppressing AKT/NF-κB mediated EMT and stemness traits, inducing apoptosis [226]. Likewise, EP restrains tumor growth and metastasis in lymphoma, as well as hepatocellular and gastric carcinomas, by downregulating the HMGB1-RAGE axis and its downstream AKT/NF-κB and ERK/STAT3/SRC signaling pathways, thereby inhibiting proliferation and provoking cell death [227,228].

Metformin has been demonstrated to directly bind the HMGB1 acidic tail and suppress HMGB1 activity and downstream inflammation mediated by NF-κB [229]. This results in downregulating the expression of MDR1, reversing cancer resistance [199], and also restraining HMGB1-induced formation of neutrophil extracellular traps (NETs), a key mechanism in the obesity-driven aggressiveness of cancer [230]. Furthermore, metformin synergizes with oxaliplatin by reducing HMGB1 expression, thereby enhancing cytotoxicity and overcoming resistance in colorectal cancer [231]. Similarly, the alkaloid berberine suppresses chemotherapy-exacerbated metastasis and synergizes with other agents by blocking the HMGB1-TLR4 axis, an effect that can be powerfully enhanced when co-delivered with doxorubicin in a biomimetic nanodrug [232,233]. Furthermore, the XIAOPI formula promotes chemosensitivity to Taxol treatment by inhibiting the CXCL1/HMGB1-mediated autophagy axis, diminishing survival pathways in breast cancer-resistant cells [137,234,235]. The natural compound triptolide was also shown to suppress breast cancer growth by directly reducing HMGB1 expression and secretion, thereby inhibiting downstream TLR4/NF-κB signaling and enhancing the efficacy of other HMGB1 inhibitors like EP [236].

Other dietary/natural agents have been also reported to directly target the HMGB1 axis to overcome chemoresistance. The alkaloid lycorine restores sensitivity to the proteasome inhibitor bortezomib in multiple myeloma by promoting the proteasomal degradation of HMGB1, which in turn inhibits pro-survival autophagy by disrupting the MEK/ERK signaling axis [237]. Agents including quercetin and aloin have been shown to downregulate HMGB1 expression, inhibit its release, and block downstream RAGE/TLR4 signaling, thereby suppressing pro-survival autophagy and NF-κB activity to restore apoptotic sensitivity [238,239,240]. Polyphenols such as EGCG promote autophagic degradation of intracellular HMGB1 and reduce extracellular HMGB1 release, thereby lowering HMGB1-mediated inflammasome activation and tumor-promoting inflammation [241,242,243]. However, the specific impact of EGCG-mediated HMGB1 inhibition in the context of reversing established cancer therapy resistance remains largely unexplored, presenting a compelling avenue for future investigation. These small molecules are attractive because many are orally bioavailable and can be repositioned rapidly, although pharmacokinetics and specificity remain important translational hurdles.

Pharmacological inhibition of HMGB1 by CDK4/6 inhibitors has been also shown to present a powerful tool to overcome breast cancer resistance to tamoxifen by disrupting the TLR4-NF-κB pathway [244]. Noteworthy, HMGB1 elevated expression in breast cancer is associated with a poorer response, though it also predicts greater benefit from CDK4/6 inhibitors, establishing HMGB1 not only as a resistance target but also as a predictive biomarker [244]. A complementary strategy is to combine HMGB1 targeting with agents that provoke immunogenic cell death (ICD) or immune checkpoint blockade. Emerging evidence also suggests that HMGB1 contributes to resistance against monoclonal antibody–based immunotherapies, such as anti-CD31 treatment in neuroblastoma, where HMGB1-driven signaling undermines therapeutic efficacy [245]. Certain chemotherapeutic formulations and nanocarriers have been engineered to trigger controlled HMGB1 release as part of an ICD program. For example, doxorubicin micelles induce ICD while concurrently downregulating c-Met, thereby enhancing antitumor immunity in breast cancer [246]. 

Finally, a few emerging agents are being evaluated in clinical trials. SB17170 (and its active metabolite SB1703) is a small molecule that selectively binds HMGB1 and, in immunocompetent syngeneic models, reduces circulating HMGB1 and tumor-associated MDSCs while enhancing T-cell infiltration and checkpoint blockade responses [247]. Dociparstat (CX-01) is a heparin derivative that binds and inhibits HMGB1. In a clinical trial for patients with hypomethylating agent-refractory AML, its combination with azacitidine proved feasible and yielded a composite response rate of 20%, demonstrating preliminary efficacy of HMGB1 therapeutic targeting in this high-risk population [248]. Although clinical HMGB1 targeting is in its infancy, these early candidates demonstrate the translational plausibility of HMGB1-directed sensitization strategies.

Collectively, these findings underscore that targeting HMGB1 through various tools can effectively dismantles the node integrating critical resistance pathways and resensitizes refractory cancer cells to various therapeutic interventions. A summary of such combinatorial therapeutic approaches involving HMGB1 modulation to overcome cancer therapy resistance to diverse anti-cancer therapies are summarized in Table 2.

## 7. Conclusions, Open Questions, and Future Perspectives

HMGB1 has emerged as a critical regulator of therapeutic resilience, acting at the intersection of genome integrity, stress adaptation, cell death control, and inflammatory signaling. Through its multifaceted roles as a nuclear DNA chaperone, cytoplasmic autophagy facilitator, and potent extracellular DAMP, HMGB1 orchestrates a broad spectrum of resistance mechanisms that collectively enable cancer cells to withstand chemotherapy, radiotherapy, targeted therapies, immunotherapy, and apoptosis-inducing agents. This remarkable functional versatility positions HMGB1 as a central node in the landscape of cancer therapy resistance.

At the nuclear level, HMGB1 promotes efficient DNA-damage repair by stabilizing nucleosomes, modulating chromatin accessibility, and facilitating BER, NER, and DSB repair, thereby mitigating genotoxic lethality. In the cytoplasm, HMGB1 interacts with Beclin-1 to drive robust autophagic flux, sustaining metabolic and redox homeostasis, preventing apoptosis, and enhancing survival under therapeutic stress. This HMGB1-cytoprotective autophagy axis further contributes to the induction of multidrug-resistance transporters, metastatic competence, and tumor recurrence [64,249]. HMGB1 additionally functions as a broad suppressor of apoptosis and other regulated cell death pathways by modulating ERK, NF-κB, c-Myc, redox balance, and GPX4 activity. Its influence on pyroptosis and ferroptosis determines whether these pathways promote antitumor immunity or resistance. Extracellular HMGB1 reinforces these programs by activating RAGE, TLR4, and TNFR1 signaling, thereby driving NF-κB/STAT3-dependent transcription of ABCB1/P-gp, ABCG2, and MRP family transporters, establishing inflammatory, pro-survival feedback loops that support tumor repopulation and therapeutic escape.

Notably, These HMGB1 tumor promoting effects are highly context-dependent, as it exhibits dual roles in cancer and can even function as a tumor suppressor in some contexts, which is largely determined by its localization, redox state, PTMs, interacting partners, and the temporal stage of tumor evolution. For instance, nuclear HMGB1 can preserve genomic stability and prevent early malignant transformation, while its stabilization of DNA-repair pathways in established cancers may instead facilitate therapeutic tolerance [250]. Similarly, extracellular HMGB1 can elicit antitumor immune responses under acute stress yet promote chronic inflammation, stromal activation, and survival signaling in persistent or therapy-selected tumors. This intrinsic duality underscores the complexity of targeting HMGB1 therapeutically; therefore, interventions must be calibrated to inhibit pathological HMGB1 signaling without compromising its physiological roles in genome maintenance and tissue homeostasis.

An important but underexplored layer of HMGB1-driven therapy resistance arises from its post-translational modifications, which dictate its stability, localization, and engagement of specific resistance pathways. Acetylation and poly-ADP-ribosylation, as well as being in a disulfide form, reduce its DNA binding and promote nuclear export, cytoplasmic accumulation and secretion, thereby enhancing pro-survival autophagy and DAMP-mediated signaling while inhibiting apoptosis [68,69,122], whereas its O-GlcNAcylation reduces the repair functions in response to genotoxic therapy [51]. Recent studies also highlight that ubiquitination can directly modulate HMGB1 stability to sustain drug-resistant phenotypes; for example, the deubiquitylase USP12 preserves HMGB1 from proteolysis by removing ubiquitin chains, thereby driving autophagy-dependent bortezomib resistance in multiple myeloma [251]. Similarly, the lncRNA MALAT1 protects HMGB1 from ubiquitin-mediated degradation, elevating its levels to promote autophagy and suppress apoptosis in MM cells [147]. Despite these advances, a systematic understanding of how distinct PTMs channel HMGB1 into specific resistance modules, DNA repair, autophagy, DAMP signaling, or ferroptosis suppression, remains lacking. Elucidating these modification-dependent “functional states” of HMGB1 represents an important future direction with strong potential for precision, PTM-selective therapeutic strategies.

Looking forward, despite intense interest, multiple unresolved questions continue to limit the translational progress of HMGB1-targeted therapeutics. First, mechanistic ambiguities remain, such as HMGB1’s paradoxical roles in NER, or the opposing immunological outcomes of distinct HMGB1 redox isoforms during ferroptosis. Addressing these contradictions will require experimental systems capable of isolating redox-specific or post-translationally modified HMGB1 variants. In addition, a deeper understanding of the spatial and temporal dynamics of HMGB1 release, particularly during therapy-induced tumor cell death, will also be essential for identifying therapeutic windows and designing rational combination regimens.

Therapeutically, a wide range of peptides, antibodies, natural compounds, and repurposed drugs have demonstrated preclinical activity in suppressing HMGB1 release or signaling, restoring sensitivity to anticancer therapies. However, no highly specific HMGB1 inhibitor has been developed to date. Current modulators act indirectly or lack high HMGB1 selectivity, increasing the risk of off-target effects. Given HMGB1’s essential roles in chromatin architecture, DNA repair, and tissue homeostasis, systemic inhibition may also produce toxicity in normal proliferative tissues. These limitations underscore that developing next-generation agents that selectively target pathological HMGB1 isoforms or redox states while demonstrating suitable pharmacokinetic profiles remains a major unmet need. In this context, nanoplatforms that pair HMGB1 blockade with ferroptosis induction, chemotherapy, or immune activation represent a promising avenue, although most remain at early translational stages. Moreover, targeted protein-modifying technologies such as proteolysis targeting chimera (PROTACs) and deubiquitinase-targeting chimeras (DUBTACs) may offer a next-generation strategy for dismantling HMGB1-centered resistance networks with greater specificity [252,253,254,255,256].

A parallel priority is the development of biomarker frameworks to guide patient selection and therapeutic monitoring. In this contexts, advances in spatial transcriptomics, proteomics, and single-cell multi-omics will be crucial for mapping HMGB1 activity across tumor niches, persister populations, and therapy-induced microenvironments. Equally important are diagnostic tools capable of assessing HMGB1 abundance, redox isoforms, compartmental localization, and secretion dynamics. These measurements will be indispensable for correlating HMGB1 states with therapy resistance, establishing on-target pharmacodynamic markers, and enabling biomarker-driven therapeutic strategies.

Ultimately, HMGB1-centered therapeutics will require a strategic shift from broad inhibition toward precise, context-dependent modulation that considers HMGB1’s biochemical states, cellular localization, and microenvironmental cues. In our opinion, HMGB1 modulation is most likely to achieve its greatest clinical benefit not as a monotherapy but as a sensitizing strategy within rational combination regimens that suppress pre-emptive stress responses or recondition the tumor immune microenvironment. Accumulating evidence support this notion; HMGB1 blockade enhances the efficacy of diverse targeted and immune therapies, including EGFR pathway inhibition through the suppression of PGRMC1-mediated EGFR activation, KRAS inhibitor resistance by suppressing pro-survival autophagy, and immune checkpoint blockade by increasing T-cell infiltration and improving anti-PD-1/PD-L1 responses [178,181,222]. Additional preclinical findings demonstrating re-sensitization to chemo- and radio-therapies by hindering DNA repair, MPR expression and potentiating cell death further underscore that HMGB1 inhibition can dismantle key resistance circuits that limit targeted, cytotoxic, and immunotherapeutic efficacy. As the field advances, the integration of systems biology, high-resolution proteomics, spatial and single-cell profiling, and next-generation therapeutic engineering [257,258,259,260] holds promise for translating HMGB1-centered interventions into clinically meaningful improvements for patients with aggressive, treatment-refractory malignancies.

## Figures and Tables

**Figure 1 ijms-26-12010-f001:**
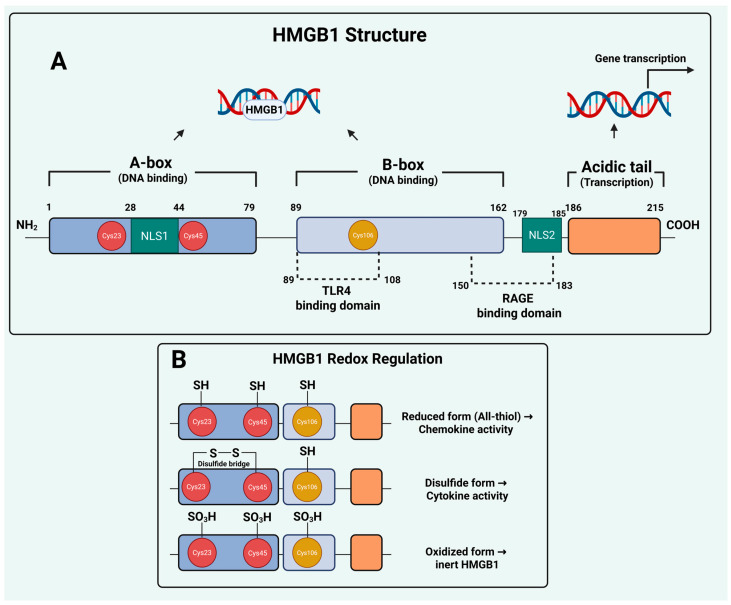
HMGB1 Structure and Redox Regulation. (**A**) Domain architecture and functional motifs of HMGB1. The DNA-binding A- and B-boxes facilitate chromatin interactions, whereas the C-terminal acidic tail modulates transcription. Localization is controlled by NLS1 and NLS2, and three conserved cysteines (C23, C45, C106) serve as redox switches that dictate extracellular behavior. (**B**) Extracellular HMGB1 signaling is governed by redox state. The reduced form exhibits chemotactic activity, the disulfide-linked form triggers pro-inflammatory cytokine release via TLR4, and the fully oxidized sulfonyl form is biologically inactive. Created in BioRender. Kachkin, D. (2025) https://BioRender.com/je4h3x8 (accessed on 1 December 2025).

**Figure 2 ijms-26-12010-f002:**
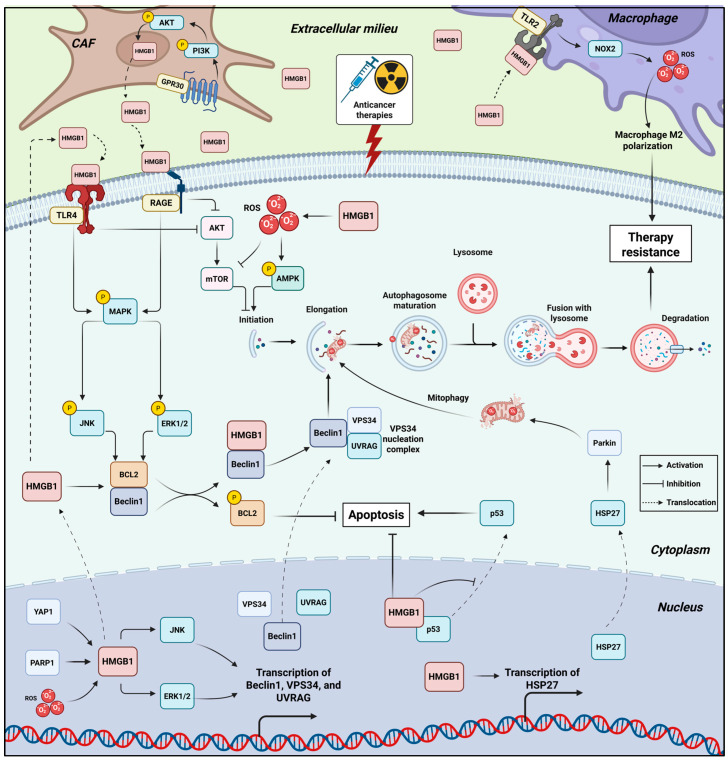
HMGB1 as a Central Orchestrator of Pro-Survival Autophagy in Cancer Therapy Resistance. This schematic illustrates the compartment-specific mechanisms by which HMGB1 activates pro-survival autophagy to confer resistance against anticancer therapies. Anticancer therapies loosen HMGB1’s interaction with chromatin, whereby nuclear HMGB1 promotes transcription of key autophagy genes Beclin1, VPS34, and UVRAG by enhancing JNK/ERK activity. HMGB1 also promotes HSP27 transcription, which facilitates mitophagy by activating Parkin. HMGB1 also restrains p53 translocation, thereby impairing apoptosis induction. Therapy-induced stress induce ROS, PARP1 and YAP activation, all trigger HMGB1’s translocation to the cytoplasm. Cytosolic HMGB1 binds Beclin-1, displacing Bcl-2 to initiate autophagosome formation, a process reinforced by ERK/JNK-mediated Bcl-2 phosphorylation. Beclin1 promotes autophagic flux while Bcl2 suppresses apoptosis. HMGB1 promotes the ROS/AMPK/mTOR pathway to further drive autophagic flux. Extracellularly, released HMGB1 in the TME acts as a DAMP, engaging receptors (RAGE, TLR4) on cancer cells to sustain pro-survival autophagy via activating MEK/ERK and inhibiting AKT/mTOR. In addition, released HMGB1 from tumor cells binds TLR2 on macrophages, activating NOX2/ROS axis that polarizes macrophages to an M2 pro-tumor phenotype, establishing a resistance-promoting niche. Chemotherapy also activates GRP30/PI3K/AKT pathway in CAFs, which promotes additional release of HMGB1 that signal through RAGE and TLR4 on tumor cells, further promoting pathways that converge to enhance autophagic flux, suppress apoptosis, and establish a robust, multi-faceted therapy-resistant state. Created in BioRender. Kachkin, D. (2025) https://BioRender.com/3p1caxx (accessed on 1 December 2025).

**Figure 3 ijms-26-12010-f003:**
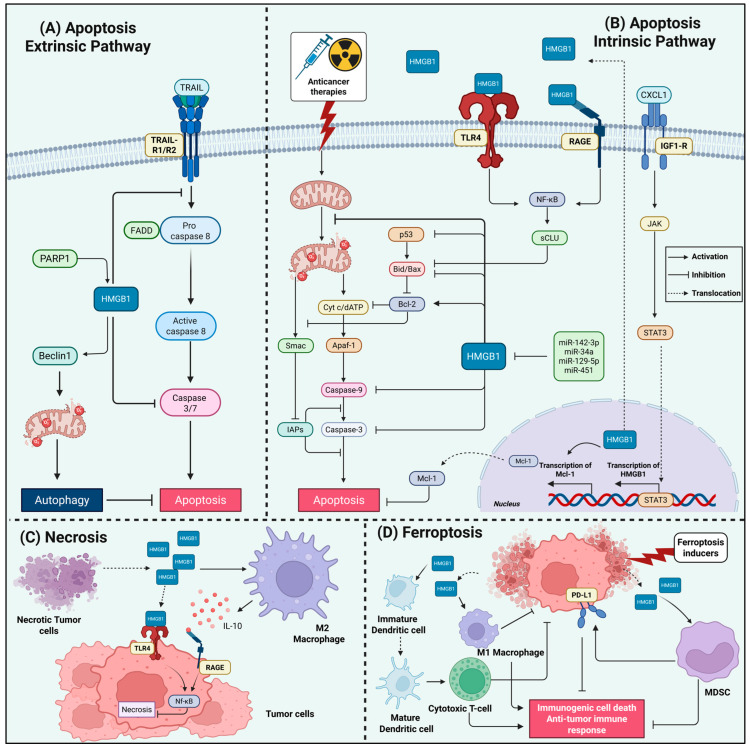
HMGB1 Interplay with Cell Death Pathways in Therapy-Resistant Cancers. This figure illustrates the mechanisms by which HMGB1 interferes with extrinsic and intrinsic apoptosis, necrosis, and ferroptosis, implicating cancer therapy resistance. (**A**) In extrinsic apoptosis pathway, HMGB1 protects against death receptor-mediated apoptosis (CD95/TRAIL) by suppressing caspase-8 activation. Under TRAIL induction, PARP1 promotes HMGB1 activity in the cytoplasm that stimulates autophagy via Beclin-1 binding, which suppresses apoptosis by diverting signaling away from caspase-3 activation. (**B**) In intrinsic apoptosis, HMGB1 suppresses mitochondrial apoptosis through multiple mechanisms: (1) Upregulation of anti-apoptotic Bcl-2 and Mcl-1 with concurrent suppression of Bax; (2) Inhibition of caspase-3/9 activation; (3) Functional inhibition of p53; (4) Released extracellular HMGB1 engages TLR4/RAGE to activate NF-κB, inducing secretory clusterin (sCLU) that sequesters Bax; (5) Macrophage-derived CXCL1 activates IGF1R/STAT3 signaling to transcriptionally upregulate HMGB1. However, multiple tumor-suppressive microRNAs (miR-142-3p, -34a, -129-5p, -451) directly target HMGB1 to restore apoptotic sensitivity. (**C**) Necrotic death augments HMGB1 release, which paradoxically functions as a survival signal that suppresses further necroptosis. HMGB1 triggers TLR4-dependent NF-κB activation, inhibiting necroptosis in adjacent tumor cells and creating a self-limiting death process. Furthermore, released HMGB1 recruits M2-like macrophages and establishes an IL-10-rich immunosuppressive milieu through RAGE signaling, facilitating tumor progression. (**D**) During ferroptosis, HMGB1 exhibits dual context-dependent roles. Ferroptotic HMGB1 release acts as a DAMP that can either: (1) promote immunosuppression by recruiting MDSCs and upregulating PD-L1 expression, fostering immune escape; or (2) enhance immunogenic cell death (ICD) by promoting dendritic cell maturation, antigen presentation, and CD8^+^ T-cell priming. Released HMGB1 can also activate pro-inflammatory M1-like macrophage polarization via RAGE binding, augmenting anti-tumor immunity. Created in BioRender. Kachkin, D. (2025) https://BioRender.com/i1en666 (accessed on 1 December 2025).

**Table 1 ijms-26-12010-t001:** Summary of HMGB1 Functions in DNA Repair Pathways and Their Implications for Therapy Resistance.

DNA Repair Pathway	Key HMGB1 Mechanisms	Cancer Context/Model	Therapy Resistance Implications	Key References
**Double-Strand Break (DSB) Repair–HR and NHEJ**	- Enhances HR and NHEJ efficiency - Promotes PARP1 expression - Interacts with Ku70/Ku80 to recruit DNA-PKcs - Supports ATM/ATR activation	ESCC, breast cancer, bladder cancer, MM, NPC	- Radioresistance (via enhancing DNA repair) - Cisplatin resistance (via Ku70 binding) - Resistance to dexamethasone via HR + autophagy	[31,32,34,35,36,37,38,39]
**Nucleotide Excision Repair (NER)**	- Cooperates with XPA–RPA and XPC–RAD23B - Facilitates recognition and excision of platinum adducts and ICLs - Redox-dependent modulation of cisplatin-DNA binding - PTMs (acetylation, phosphorylation, O-GlcNAcylation) regulate DNA binding and lesion repair	UVC models, cisplatin-treated cells	- Promotes cisplatin resistance when enhancing NER - Can sensitize cells when strongly binding cisplatin adducts (redox-dependent)	[40,41,42,43,44,45,46,47,48,49]
**Mismatch Repair (MMR)**	- Interacts with MLH1 and MSH2 - Stimulates EXO1 activity - Bridges MMR–NER complexes for ICL processing	In vitro reconstituted MMR, MEFs	- No direct evidence yet linking HMGB1–MMR to resistance - Suggests a potential but unproven role in ICL-based drug responses	[52,53,54,55,56]
**Base Excision Repair (BER)**	- Inhibits SN-BER by competing with Pol β - Stimulates LP-BER via APE1 and FEN1 - Modulates repair of oxidative lesions	Cell-free systems, BER enzyme assays	- No direct evidence for therapy-resistance relevance to date - Potential implications for oxidative stress-based therapy	[30,57,58,59]
**Overall Impact on Genome Maintenance**	- Coordinates chromatin remodeling and repair complex assembly - Accelerates damage resolution under stress	Multiple cancer types	- Global radio- and chemoresistance enhancement - HMGB1 deletion consistently sensitizes tumors to DSB-inducing therapies	[30,31,35,60]

**Table 2 ijms-26-12010-t002:** Combined therapeutic strategies involving HMGB1 targeting to overcome cancer therapy resistance.

HMGB1-Targeting Strategy	Co-Therapy/ Drug Class	Cancer Type	Mechanistic Rationale	Observed Outcome	Reference(s)
**Anti-HMGB1 antibodies**	Docetaxel, Doxorubicin, Paclitaxel, Adriamycin, Vincristine, Cytosine arabinoside, Arsenic trioxide	Leukemia, Prostate and breast cancer models	Neutralization of HMGB1 reduces autophagy and NF-κB/c-Myc, and MEK/ERK activation	Increases apoptosis overcomes resistance to various chemotherapeutic drugs	[94,103,125,135]
**Glycyrrhizin (HMGB1 binder; NHEJ disruption)**	Cisplatin/Ionizing radiation	Nasopharyngeal carcinoma	Glycyrrhizin inhibits HMGB1–Ku70 interaction → impairs NHEJ and HMGB1-mediated survival signaling	Restores cisplatin and radiotherapy sensitivity; increases DNA damage (γH2AX); reduces cell viability.	[37]
**Glycyrrhizin (HMGB1 inhibitor; PGRMC1 effect)**	EGFR-pathway inhibitors/EGFR signaling	NSCLC	GL inhibits HMGB1 interactions and also binds PGRMC1, indirectly restraining EGFR activation, providing rationale for combining GL with EGFR inhibitors	Suppressed EGFR signaling and restored sensitivity to chemotherapy/targeted agents in preclinical models.	[222]
**Glycyrrhizic acid-loaded nanoplatform**	Anti-PD-L1 immunotherapy (immune checkpoint blockade)	Pancreatic cancer	Blocks HMGB1 release	Enhances ferroptosis-induced anti-tumor immunity, reduces MDSC recruitment, sensitizes cancer cells to immunotherapy	[181]
**Glycyrrhizin**	Paclitaxel	Pancreatic cancer	Suppresses HMGB1-mediated c-MYC activation	Potentiates apoptosis, restores chemosensitivity	[125]
**HMGB1 Box A peptide/Box A gene therapy**	Cisplatin	Cisplatin-resistant NSCLC	Box A competes with full-length HMGB1 (extracellular antagonism) and intracellular Box A induces DSBs/perturbs HMGB1 localization → impairs DNA repair and autophagy	Re-sensitization to cisplatin; increased DNA damage and apoptosis.	[216,217]
**HBHP peptide (binds HMGB1 A box)**	Oxaliplatin, **Etoposide**	Lung and colorectal cancer	disrupts the HMGB1-Hsp70 DAMP complex, inhibits PGE2-induced inflammation and autophagy	Inhibits repopulation of residual cancer cells following chemotherapy	[203]
**SB17170 (HMGB1 inhibitor/prodrug)**	Anti-PD-1/PD-L1 immune checkpoint blockade	Immunocompetent syngeneic tumor models	SB17170 reduces extracellular HMGB1, lowers MDSCs, increases T-cell infiltration → enhances responsiveness to checkpoint blockade	Enhanced checkpoint blockade efficacy and improved T-cell infiltration in preclinical models.	[247]
**Dociparstat sodium (CX-01; HMGB1 binder)**	Azacitidine (hypomethylating agent)	AML (clinical combination pilot/trial)	CX-01 binds/inhibits HMGB1 and alters microenvironmental inflammatory signaling → may potentiate azacitidine activity	Feasible combination with signals of clinical activity in refractory patients.	[248]
**Metformin (HMGB1 downregulation)**	Oxaliplatin (platinum chemotherapy)	Colorectal cancer (DLD-1 cells, preclinical)	Metformin reduces HMGB1 expression/release and associated NF-κB activity → reduces autophagy and survival pathways	Synergistic cytotoxicity with oxaliplatin and reduced HMGB1 levels.	[231]
**Berberine (HMGB1-TLR4 axis inhibitor) delivered with doxorubicin in nanocarrier**	Doxorubicin (chemotherapy)	Breast cancer (preclinical, biomimetic nanodrug)	Berberine + doxorubicin co-delivery blocks HMGB1-TLR4 signaling, enhancing chemo efficacy	Improved chemosensitivity and reduced metastasis in preclinical models.	[232,233]
**XIAOPI formula**	Paclitaxel (taxol)	Breast cancer models	Inhibits CXCL1/HMGB1-mediated autophagy → sensitizes resistant cells to taxane	Restored sensitivity to paclitaxel in resistant models.	[137,234]
**Ethyl pyruvate (inhibits HMGB1 release/translocation)**	Cisplatin/Chemotherapy	Cervical, lung, prostate, lymphoma (preclinical)	EP blocks HMGB1 translocation/release → reduces pro-survival autophagy and NF-κB/AKT signaling	Reversed cisplatin resistance, switched necrotic death to apoptosis, improved chemo response.	[224,225,226,227,228]
**Lucidone**	Gemcitabine	Pancreatic cancer	Suppresses HMGB1-mediated autophagic flux and MDR1 expression;	Promoting apoptosis and overcoming resistance to gemcitabine	[98]
**Lycorine (promotes proteasomal HMGB1 degradation)**	Bortezomib (proteasome inhibitor)	Multiple myeloma (preclinical)	Promotes HMGB1 degradation and inhibits autophagy	Restored sensitivity to bortezomib.	[237]

## Data Availability

No new data were created or analyzed in this study. Data sharing is not applicable to this article.

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
