# Peer review of "HMGB1: A Central Node in Cancer Therapy Resistance"

_ijms, 2025, doi:10.3390/ijms262412010_

Round 1
Reviewer 1 Report
Comments and Suggestions for Authors
The review touches important matter of cancer resistance and the role of HMGB1 in this aspect. The authors state clearly in their introduction their focus and provide a reader with the roadmap thorugh the article - DNA rapair, autophagy, cell death mechanisms, MRPs, and therapeutic strategies.
One general comment I have is that the authors have clearly put a great deal of effort into the manuscript, and it contains a substantial amount of information. My remarks therefore concern primarily the structure, as the main work needed at this stage relates to refining the text itself. The text is at times difficult to read, and there are frequent repetitions of information that is presented as if it were new. I kindly ask the authors to review the manuscript with this in mind—consider adding a figure or inserting a table where appropriate, and rewriting some sections so that they concisely highlight the information that is most important. Apart from that, it is a good piece of work, and I congratulate the authors on the effort they have put into it.
Since there are so many proteins and pathways, there should be section with the full form of abbreviations. Many of them are not expanded within the text.
Abstract
In general, abbreviations should either be expanded or avoided in abstracts.
Introduction:
I believe this section can be re-written to make it more concise.
-Line 90 it says "depending on its redox state,..." and than, just a section below (line 97) "a critical determinant of HMGB1 function is its redox state...". The authors refer to redox state a lot, so this fragment should be comprehensible, short and clear for the reader that this changes the game.
-The information about extracellular HMGB1 being a DAMP or alarmin should be underlined and clear, instead it appears several times - line 89, than a section 111-122. Maybe the authors can organize the introduction better?
-Lines 58-72, maybe it should be just summarized with a graph. As there are no references to this chemical structures later on.
-There are multiple review articles regarding HMGB1's role in cancer and its environment, please write a section, citing all the relevant reviews to respect the authors, and explain to the reader what is the novelty in your review. Why should one read it?
Body
-Please either summarize section number 2 about DNA repair with graph or table.
-Lines 403-406 - main function of DAMP is to alarm the immune system, so I would say that streightforward activation of pro-survival pathways is not "prototypical DAMP".
-Autophagy seems very complex. Maybe the authors could create a graph summarizing the nuclear localization and/or redox state and its effect on the cell? A simple, schematic picture.
-Can the authors add a sentence regarding relation between senescence and HMGB1? Senescence is quite common way to survive the stress and lines 599-605 explain the intrplay between p53 and HMGB1. Since it also affects DNA repair mechanisms, it must affect also senescence induction mechanisms within the cell?
-I wonder if the authors could also discuss the notion that nuclear HMGB1 can have a dual role in cancer, acting as an oncogene and a tumor suppressor. It was expanded in this review, for instance: 10.3389/fonc.2024.1384109
-The last section (nr 6) contains therapeutic strategies. The authors mention the three approaches. Please, add a table showing the literature examples of those approaches.
Future Perspectives
-Can you identify any limitations of HMGB1-focused therapies, considering its broad role in cellular biology?
Reviewer 2 Report
Comments and Suggestions for Authors
This is a timely and comprehensive review that convincingly positions HMGB1 as a master regulator of multifaceted therapy resistance in cancer. The manuscript synthesizes a vast body of literature into a coherent model, highlighting compartment-specific functions of HMGB1. The scope is ambitious and generally well-executed, with clear graphical abstracts aiding comprehension. The topic is of significant interest to the readership of International Journal of Molecular Sciences. However, several areas require refinement to meet the journal's high standards for clarity, depth, and formal presentation.
- The section "HMGB1 and Cell Death Mechanisms" in the article lacks a direct association between the discussion of ferroptosis and pyroptosis with clinical drug resistance, remaining only at the level of basic mechanisms. The paper mentions the dual regulation of ferroptosis by HMGB1 but fails to clarify how this regulation affects the resistance to specific chemotherapeutic drugs (such as sorafenib and erastin), nor does it integrate recent preclinical data on ferroptosis inducers combined with HMGB1 inhibitors. This results in an insufficient connection between the section and the core theme of therapeutic resistance.
- There is no dedicated section on "HMGB1 as a drug resistance biomarker," which overlooks its potential in clinical prognosis assessment and treatment response prediction, failing to fully explore the clinical value of the research focus. Recent studies have confirmed that HMGB1 expression levels are closely related to the response rates of chemotherapy and immunotherapy. However, the paper only briefly mentions this in the discussion section and does not systematically sort out the clinical evidence of HMGB1 as a biomarker (such as the correlation between serum HMGB1 levels and the efficacy of PD-1 inhibitors), which weakens the practical value of the research focus.
- Section 3.3 (Extracellular HMGB1) and Section 5 (HMGB1 as a Molecular Driver of Multidrug Resistance) of the review discuss how extracellular HMGB1, as a DAMP molecule, promotes resistance and the role of exosomes in the microenvironment.
- The insufficient elaboration of cross-regulatory mechanisms between various chapters, such as the lack of integrated analysis on the coordinated regulation of DNA repair and autophagy, and the association between cell death and multidrug resistance (MDR). The paper separately discusses the role of HMGB1 in pathways such as DNA repair and autophagy, but fails to clarify the cross logic of "HMGB1 enhances DNA repair while activating autophagy" and does not explain how the escape of cell death promotes the expression of MDR transporters, resulting in the framework lacking an overall perspective of multi-pathway synergistic drug resistance.
- Section 4.2 of the review (HMGB1 and Necrosis/Necroptosis) describes the regulatory role of HMGB1 in necrosis and its association with NLRP3. This literature (PMID: 32770173) links SIRT3 deficiency, necrosis, and NLRP3 activation, which is consistent with the discussion of necrosis in the literature. It can be inserted into the paragraph describing "HMGB1 release can drive pro-tumor effects via necroptosis" to enrich the mechanistic details of HMGB1 and necrotic signals (such as TLR4/NF-κB).
- The core innovation point, "the multi-dimensional drug resistance regulation of HMGB1," lacks an analysis of the functional differentiation of its different modified forms (acetylation, phosphorylation, O-GlcNAcylation). The paper mentions that post-translational modifications of HMGB1 are involved in localization regulation, but it does not systematically compare the specific effects of different modifications on drug resistance pathways (such as acetylation promoting autophagy and O-GlcNAcylation affecting DNA repair). However, modification-dependent targeting is an innovative direction in the field in recent years, and this omission reduces the technical depth of the innovation point.
- The review provides a detailed analysis of the dual roles of HMGB1 in ferroptosis (such as promoting GPX4 inhibition or antioxidant effects).
- The discussion section does not explicitly address the combined therapeutic strategies of HMGB1 targeting with other drug-resistant targets (such as EGFR and KRAS), and fails to respond to the clinical demand for "combined targeting". Explanation: In clinical practice, targeting a single target is prone to drug resistance, but the paper does not explore the combined application effects of HMGB1 inhibitors with EGFR inhibitors and immune checkpoint inhibitors, nor does it cite relevant preclinical combined therapy data, resulting in a lack of constructive ideas for clinical application in the discussion.
- Section 4.4 (HMGB1 in Pyroptosis) of the review focuses on discussing the role of HMGB1 in pyroptosis, and Section 6 (Therapeutic Strategies) mentions HMGB1 targeting strategies.
- The formatting of the references in the article is inconsistent. For example, some journal names are in full while others are abbreviated; page numbers are incomplete or missing; what does the attribute "Internet" for journal articles describe; and some lack information on volume and issue, among other issues.
- Section 3 of the review (HMGB1/Autophagy Interplay) focuses on discussing the core role of HMGB1 in autophagy, including transcriptional regulation and interaction with Beclin-1.
Round 2
Reviewer 1 Report
Comments and Suggestions for Authors
The authors have introduced changes following the reviewer's suggestions.
Reviewer 2 Report
Comments and Suggestions for Authors
Everything looks good.